# Uncertainty-Aware Systems for Human-AI Collaboration

**Vasco Pearson**                                                        *vasco.pearson@feedzai.com*
*Feedzai*

**Jean V. Alves**                                                          *jean.alves@feedzai.com*
*Feedzai*

**Jacopo Bono**                                                          *jacopo.bono@feedzai.com*
*Feedzai*

**Mário A.T. Figueiredo**                                  *mario.figueiredo@tecnico.ulisboa.pt*
*Instituto Superior Técnico, Universidade de Lisboa*
*Instituto de Telecomunicações*

**Pedro Bizarro**                                                        *pedro.bizarro@feedzai.com*
*Feedzai*

**Reviewed on OpenReview:** *https://openreview.net/forum?id=PiRYCyNBqQ*

## Abstract

*Learning to defer* (**L2D**) algorithms improve human-AI collaboration (**HAIC**) by deferring decisions to human experts when they are more likely to be correct than the AI model. This framework hinges on machine learning (**ML**) models' ability to assess their own certainty and that of human experts. L2D struggles in dynamic environments, where distribution shifts impair deferral. We argue that robust HAIC in dynamic environments requires uncertainty-driven policy switching rather than reliance on a single deferral strategy. To operationalize this principle, we introduce two uncertainty-aware approaches that estimate epistemic uncertainty to guide the deferral policy choice. Both methods are the first uncertainty-aware approaches for HAIC that also address limitations of L2D systems including cost-sensitive scenarios, limited human predictions, and capacity constraints. Empirical evaluation in fraud detection shows both approaches outperform state-of-the-art baselines while improving calibration and supporting real-world adoption.

## 1 Introduction

While artificial intelligence (**AI**) and machine learning (**ML**) models rival human expert performance and efficiency in various domains, including finance (Khandani et al., 2010; Awoyemi et al., 2017), criminal justice (Goel et al., 2021), and healthcare (Gulshan et al., 2016; Somanchi et al., 2015; Esteva et al., 2017), their use in high-stakes settings still faces challenges. Their reliance on training data constrains their scope and they often struggle to generalize under dynamic conditions, such as in adversarial settings in finance (Perdomo et al., 2020; Lunghi et al., 2023), or under concept drift (Gama et al., 2014). Furthermore, their predictions often lack transparency, posing difficulties to interpretability and trust (Saeed & Omlin, 2023; Gohel et al., 2021). As a result, their applicability in automated decision-making systems remains constrained.

In contrast, humans can access external information, adapt to changing environments, and provide causal reasoning and explanations for decisions (Gopnik & Wellman, 2012). These attributes make human experts indispensable in scenarios where interpretability, adaptability, and broader contextual awareness are required. Consequently, combining the complementary strengths of humans and ML models through human-AI collaboration (**HAIC**) systems has emerged as a promising approach to address the limitations of fully automated systems (Dellermann et al., 2019; De-Arteaga et al., 2020).

Optimizing the allocation of decisions to humans or AI is a major challenge in HAIC. The simplest approach, rejection learning (**ReL**) (Chow, 1970; Hellman, 1970; Cortes et al., 2016; Geifman & El-Yaniv, 2017), addresses this by deferring to human experts instances with high model uncertainty. However, an optimal solution would involve taking human certainty into consideration, assigning instances to humans only when they are likely to outperform the model. Learning to defer (**L2D**) aims to achieve this by estimating the performance of both the model and the human decision-maker, thereby enabling a better allocation of tasks to the decision-maker most likely to succeed (Madras et al., 2018; Mozannar & Sontag, 2020; Mozannar et al., 2023; Verma & Nalisnick, 2022; Verma et al., 2023; Hemmer et al., 2022; 2023; Charusaie et al., 2022).

Although L2D improves decision-making performance, it struggles in dynamic environments. L2D methods rely on ML models to estimate the correctness of each decision-maker, whether human or AI. However, when faced with distributional changes, these models often become unreliable (Gama et al., 2014). Beyond performance degradation, a critical issue is that they can exhibit high confidence in their predictions on out-of-distribution (**OOD**) samples (Hein et al., 2019), which in an L2D system leads to suboptimal assignments. In contrast, ReL, which leverages uncertainty measures (Hendrickx et al., 2024), can detect such changes at inference time, deferring novel instances to human experts, thus ensuring robust handling of OOD data.

Unlike current ML models, humans can adapt to changing environments, a crucial advantage that HAIC systems should be designed to leverage. Ideally, one would rely on the models whenever they reliably perform well, which requires that model uncertainty be estimated in a way that allows humans to intervene when necessary. However, current L2D approaches remain limited in this respect, as they focus on obtaining calibrated performance prediction, without incorporating uncertainty estimation into the deferral process. Integrating uncertainty estimation techniques with optimal decision allocation in HAIC is nontrivial, since this combination introduces new theoretical and practical challenges and tradeoffs, which are not present when each is studied in isolation. To address this gap, we show that robust human–AI collaboration under distribution shift requires uncertainty-driven policy switching rather than reliance on a single deferral strategy (Figure 1). Instead of committing to either pure L2D or pure rejection-based deferral, HAIC systems must adapt their routing policy across uncertainty regimes. We implement this principle through two uncertainty-aware methods for HAIC and provide an empirical study of their advantages and limitations.

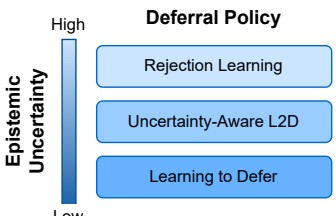

Figure 1: Uncertainty-driven policy switching in human-AI collaboration.

In addition to challenges posed by dynamic environments, L2D systems have failed to address other practical constraints (Leitão et al., 2022; Alves et al., 2024), including human work capacity limitations, reliance on extensive expert predictions for training, and limited consideration of cost-sensitive scenarios. While prior work (Hemmer et al., 2023; Tailor et al., 2024; Alves et al., 2024) has addressed some of these issues, our approach accounts for all of the aforementioned constraints, enabling robust and practical deployment.

In summary, this work makes three key contributions to HAIC:

- We demonstrate that robust human–AI collaboration under distribution shift benefits from uncertainty-driven policy switching rather than reliance on a single deferral strategy. We show that effective HAIC systems must adapt their routing policy based on epistemic uncertainty and distributional context.
- We introduce two uncertainty-aware systems that operationalize this principle: (i) an uncertainty-aware L2D framework and (ii) a hybrid system combining conformal prediction with L2D, both explicitly accounting for misclassification costs and human capacity constraints.
- We provide an extensive empirical evaluation in a realistic, cost-sensitive fraud detection setting using a benchmark dataset (Alves et al., 2025a) with synthetic expert decisions modeled after real analysts. Across variations in noise, data availability, and work capacity constraints, both systems consistently improve performance over state-of-the-art baselines.

## 2 Related Work

**ReL and uncertainty estimation.** The simplest HAIC approach in the literature is ReL (Chow, 1970; Hellman, 1970; Cortes et al., 2016; Geifman & El-Yaniv, 2017), which often involves having the model abstain from predicting in high uncertainty cases (Hendrycks & Gimpel, 2017). Hendrickx et al. (2024) categorize rejection into two types: *ambiguity rejection*, which enables the model to abstain in scenarios where the target values are inherently ambiguous, and *novelty rejection*, where the model refrains from predicting for instances that deviate significantly from the training data. These types of rejection align with different types of uncertainty: *aleatoric uncertainty*, which stems from irreducible randomness within the data (such as class overlap), thus leading to ambiguity rejection; *epistemic uncertainty*, which arises from incomplete knowledge, whether due to distribution shifts or uncertainty about the model fit to the data, leading to novelty rejection (Hüllermeier & Waegeman, 2021). Several studies have proposed methods to distinguish these types of uncertainty for applications like ReL (Senge et al., 2014) and active learning (Nguyen et al., 2019). Generative models have been proposed as an intuitive way to quantify epistemic and aleatoric uncertainty (Hechtlinger et al., 2018; Sun et al., 2023; Postels et al., 2020), by leveraging estimates of the density function $p(\boldsymbol{x})$, where $\boldsymbol{x}$ is a feature vector, to decide whether input points are located in regions of high or low density, in which the latter serves as a proxy for high epistemic uncertainty.

**Learning to defer.** The L2D framework was introduced by Madras et al. (2018) to address the shortcomings of confidence-based ReL, which does not take into account the human decision-maker's performance. Leveraging not only class labels but also predictions from the downstream human decision-maker, a classifier and a deferral mechanism are trained, meaning L2D considers both model and human errors when optimizing decisions. Mozannar & Sontag (2020) highlight that the loss function proposed by Madras et al. (2018) is not consistent, proposing a consistent surrogate loss that includes a separate class for deferral. Verma & Nalisnick (2022) critique the L2D surrogate loss developed by Mozannar & Sontag (2020) for miscalibration issues, introducing a one-vs-all (**OvA**) approach that improves calibration by training the classifier and deferral model independently. In the multi-expert setting, Keswani et al. (2021) extend L2D to assign instances to multiple experts, while Verma et al. (2023) propose a new consistent and calibrated loss, generalizing the loss of Verma & Nalisnick (2022) to handle multi-expert scenarios.

However, L2D methods face several limitations in practical deployment (Leitão et al., 2022). They lack adaptability in dynamic environments, where data distributions can evolve over time due to concept drift (Gama et al., 2014), or in performative prediction (Perdomo et al., 2020). Changes in distribution degrade the performance of the ML models used in L2D to estimate the correctness of the decision-makers, which leads to sub-optimal assignments. Moreover, these frameworks rely heavily on labeled data sourced from all experts involved, rendering them impractical in real-world scenarios due to high labeling costs. Most existing L2D methods also do not consider cost-sensitive scenarios where misclassification costs can vary. Finally, these methods often neglect human work capacity constraints at inference time.

Recent advancements in L2D research have aimed to address some of these practical constraints, particularly the challenge of learning with limited expert data (Charusaie et al., 2022; Hemmer et al., 2023; Tailor et al., 2024). Alves et al. (2024) focus on cost and capacity constraints, which are crucial for real-world applications. Despite these advances, L2D research has not focused on its limitations in dynamic environments. Since human experts adapt to changing conditions, HAIC systems can mitigate this, but doing so requires reliable uncertainty estimates to defer cases where models may fail.

**Synthetic human expert predictions.** Datasets with human predictions are scarce and collecting annotations is often prohibitively expensive. Prior work resorts to simulating human decision-makers by injecting label noise. Verma & Nalisnick (2022), Mozannar & Sontag (2020), and Charusaie et al. (2022) use CIFAR-10 (Krizhevsky, 2012) to create experts with high accuracy on some classes while deciding randomly on others, while Keswani et al. (2021) simulate racially biased experts by varying error rates across groups.

While these methods provide a simple way to introduce heterogeneity across experts, they have a major limitation: their accuracy is determined solely by class labels or a single feature, ignoring the broader set of input features that influence real expert decision-making. To the best of our knowledge, the most principled approach to synthetic expert simulation in L2D is the OpenL2D framework (Alves et al., 2025a).

This framework addresses the aforementioned limitations by generating synthetic expert predictions through instance-dependent label noise, where the probability of error is a function of instance features and additional information available to decision-makers (e.g., a risk score provided by a separate ML model (De-Arteaga et al., 2020)). This provides a more realistic approximation of human decision-making behavior. By applying this framework to a public fraud detection dataset, Alves et al. (2025b) produce the Financial Fraud Alert Review (**FiFAR**) dataset, which contains predictions from synthetic fraud analysts over 30K instances. Crucially, this dataset was constructed with the explicit goal of realistically simulating the behavioural properties (*e.g.,* interrater agreement, consistency, performance distribution) of a team of highly skilled experts. These synthetic experts are supported by decision-making literature and were modeled after real financial fraud analysts. By adopting this dataset, our empirical evaluation of L2D methods is, to the best of our knowledge, based on the most realistic set of expert decisions to date.

## 3 Uncertainty-Aware L2D

To address the limitations of L2D systems in dynamic environments, we propose an uncertainty-aware L2D framework that improves how correctness probabilities are estimated for both the classifier and human experts. Deferral decisions in L2D rely directly on these estimates: instances are assigned to the decision-maker predicted to be most likely correct. When distribution shift occurs, standard ML models can become overconfident on OOD instances (Hein et al., 2019), leading to unreliable correctness estimates and suboptimal assignment decisions. To mitigate this, we incorporate density-aware modeling (Bui & Liu, 2024) into the L2D framework (Figure 2). By using density information as a proxy for epistemic uncertainty, the system attenuates confidence in low-support regions of the feature space. This produces more reliable correctness estimates under distribution shift, thereby enabling more robust deferral decisions and improved expert modeling when labeled data is limited.

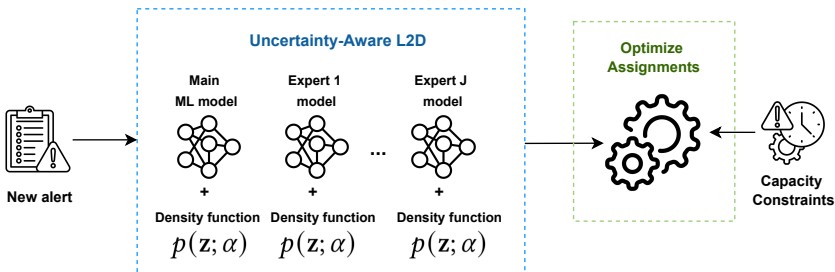

Figure 2: Uncertainty-aware L2D.

**Density-softmax.** Density-softmax (Bui & Liu, 2024) can be viewed as a data-dependent temperature scaling mechanism: logits are attenuated in low-density regions, increasing predictive entropy. This behavior qualitatively approximates epistemic variance shrinkage in Bayesian models, where uncertainty over parameters reduces confidence away from observed data. Unlike Bayesian methods, however, density-softmax achieves this effect deterministically without modifying model weights. It consists of a feature extractor, a density estimator based on normalizing flows (**NF**) (Dinh et al., 2017), and a classifier. The feature extractor $f$ maps an input $\boldsymbol{x}$ into a latent space representation $\mathbf{z}$, where the NF model estimates the log-likelihood $\log p(\mathbf{z}; \alpha)$ under the training distribution, with $\alpha$ representing the model parameters. The NF model is optimized via maximum likelihood estimation (**MLE**) and provides exact log-likelihoods. Importantly, Bui & Liu (2024) show that the density-softmax approach achieves lower test-time latency than state-of-the-art uncertainty estimation methods, making it a lightweight addition to an L2D system with minimal computational overhead. To avoid numerical issues from unbounded log-likelihoods, we scale them following Bui & Liu (2024): at test time, log-likelihoods are divided by the maximum training log-likelihood and then min–max normalized to $(0, 1]$. The scaled likelihood $p(\mathbf{z}; \alpha)$ adjusts the classifier $g$'s logits, modifying the predictive probability as follows:

$$p(y = i | \boldsymbol{x}) = \frac{\exp(p(\mathbf{z}; \alpha)\ g_i(\boldsymbol{x}))}{\sum_{j=1}^{K} \exp(p(\mathbf{z}; \alpha)\ g_j(\boldsymbol{x}))}, \tag{1}$$

where $g_i$ is the logit of classifier $g$ for class $i$. For OOD instances in low-density regions, this adjustment scales down the logits, resulting in predictive probabilities that reflect higher uncertainty and improve calibration. These calibrated probability estimates are then used in the optimization step described in Section 5.

**L2D Formulation.** Our L2D system is built using the OvA framework introduced by Verma et al. (2023), which aims to minimize the 0-1 loss defined as

$$L_{0-1}(h, r) = \mathbb{E}_{\boldsymbol{x}, \mathrm{y}, \{m_1, \ldots, m_J\}} \left[ \mathbb{I}[r(\boldsymbol{x}) = 0]\, \mathbb{I}[h(\boldsymbol{x}) \neq \mathrm{y}] + \sum_{j=1}^{J} \mathbb{I}[r(\boldsymbol{x}) = j]\, \mathbb{I}[m_j \neq \mathrm{y}] \right],$$

where $h : \mathcal{X} \to \mathcal{Y}$ is the classifier, $r : \mathcal{X} \to \{0, 1, \ldots, J\}$ is the rejector, and $m_j$ is the prediction from expert $j \in \{1, \ldots, J\}$. When $r(\boldsymbol{x}) = 0$, the classifier $h$ makes the decision, and when $r(\boldsymbol{x}) = j$, the decision is deferred to the $j$-th human expert. Verma et al. (2023) show that minimizing the 0-1 loss leads to the following Bayes optimal classifier and rejector:

$$h^*(\boldsymbol{x}) = \arg\max_{y \in \mathcal{Y}} \mathbb{P}(y \mid \boldsymbol{x}), \tag{2}$$

$$r^*(\boldsymbol{x}) = \begin{cases} 0 \text{ if } \mathbb{P}(\mathrm{y} = h^*(\boldsymbol{x}) \mid \boldsymbol{x}) > \mathbb{P}(\mathrm{y} = m_{j'} \mid \boldsymbol{x})\, \forall j' \\ \arg\max_{j \in \{1, \ldots, J\}} \mathbb{P}(\mathrm{y} = m_j \mid \boldsymbol{x}) \text{ otherwise}, \end{cases} \tag{3}$$

where $\mathbb{P}(\mathrm{y} \mid \boldsymbol{x})$ is the probability of the label under the data generating process and $\mathbb{P}(\mathrm{y} = m_j \mid \boldsymbol{x})$ is the true probability that the $j$th expert is correct.

To approximate this solution, the OvA framework constructs the classifier $h$ and rejector $r$ using $|\mathcal{Y}|$ functions $g_k : \mathcal{X} \to \mathbb{R}$, where each function $g_k$ is related to the probability of the instance belonging to class $k$, and $J$ functions $g_{\perp,j} : \mathcal{X} \to \mathbb{R}$ related to the likelihood of each expert $j \in J$ making the correct decision. These $|\mathcal{Y}| + J$ functions are combined in the OvA surrogate loss, defined as

$$\Psi_{\mathrm{OvA}}^{J} = \Phi[g_y(\boldsymbol{x})] + \sum_{y' \in \mathcal{Y}, y' \neq y} \Phi[-g_{y'}(\boldsymbol{x})] + \sum_{j=1}^{J} \Phi[-g_{\perp,j}(\boldsymbol{x})]$$

$$+ \sum_{j=1}^{J} \mathbb{I}[m_j = y](\Phi[g_{\perp,j}(\boldsymbol{x})] - \Phi[-g_{\perp,j}(\boldsymbol{x})]), \tag{4}$$

where $m_j$ represents the $j$-th expert's decision, and $\Phi : \{\pm 1\} \times \mathbb{R} \to \mathbb{R}_+$ is a strictly proper binary surrogate loss function.

Verma et al. (2023) prove that the minimizer of the pointwise inner $\Psi_{\mathrm{OvA}}^{J}$-risk is composed of the minimizers of the inner $\Phi$-risk for each binary classification problem, meaning each $g_y$ can be trained independently on its corresponding OvA binary classification task by minimizing a proper binary loss (such as the log-loss), and calibrated estimates for a given instance's probability of belonging to each class $k$ are given by $\psi^{-1}(g_y)$, where $\psi^{-1}$ is the inverse link function for the proper binary surrogate loss $\Phi$ (in the case of the log-loss, $\psi^{-1}$ is given by the sigmoid function $\sigma$). Similarly, each $g_{\perp,j}$ can be trained independently on the subset of data where predictions from expert $j$ are available, and calibrated estimates for the probability that expert $j$ will predict correctly on instance $i$ are given by $\psi^{-1}(g_{\perp,j})$.

**Aleatoric and Epistemic Uncertainty.** By using the density-softmax approach, each of the $|\mathcal{Y}| + J$ binary classifiers is paired with a density function to improve uncertainty estimation. The allocation of instances to decision-makers relies on these correctness estimates, which must be calibrated and reflect instance-specific uncertainty. Equation 1 contains both aleatoric and epistemic uncertainty. To disentangle these, the likelihood value from the density model must be separated from that of the main classifier, allowing it to represent epistemic uncertainty. In regions of high epistemic uncertainty, it is reasonable to downweigh the classifier's predictions, as the classifier relies solely on training data and may not generalize well to OOD instances. Human experts, however, can adapt, learn, and access additional information not available to machine learning models, meaning we can expect them to outperform the classifier on OOD

instances. Consequently, we use the human expert's average correctness on the training data as a proxy for their expected performance. Specifically, for instances with high epistemic uncertainty, the model blends the predicted logits of the binary classifier for expert $j$ ($\hat{g}_{\perp,j}$) with the expert's average probability of correctness on the training data ($\hat{p}_{j,\text{avg}}$), using the density score $p(\mathbf{z};\alpha)$ from the normalizing flows model ($p(\mathbf{z};\alpha) \in (0,1]$). The adjusted logits are defined as

$$p(\mathbf{z};\alpha)\hat{g}_{\perp,j} + (1 - p(\mathbf{z};\alpha))\sigma^{-1}(\hat{p}_{j,\text{avg}}), \tag{5}$$

where $\sigma$ represents the sigmoid function, and are fed into the softmax function in Equation 1. This adjustment ensures that the model reflects a more realistic estimate of expert performance in high epistemic uncertainty instances, balancing the model's own estimates with the historical expert performance.

# 4 Conformal Prediction for HAIC

In this section, we introduce our second approach to uncertainty-aware HAIC, which follows a different design principle. Whereas the previous approach focuses on providing calibrated estimates of correctness to improve assignment within an L2D framework, here we explicitly separate in-distribution and OOD instances and apply different deferral strategies to each. We hypothesize that, while L2D methods perform well on familiar, in-distribution cases, these should avoid handling instances lying outside the training distribution. For such OOD cases, the models' probability estimates can be poorly calibrated, resulting in suboptimal decisions. In these scenarios, human experts are better suited to make predictions, as they are able to generalize and adapt by leveraging broader contextual knowledge. Furthermore, in high-stakes environments, it may be preferable, or even required, that OOD instances be reviewed by human experts to ensure accountability and explainability. To meet these requirements, we propose a system that defers OOD instances to human experts through ReL, while relying on L2D for in-distribution cases (Figure 3).

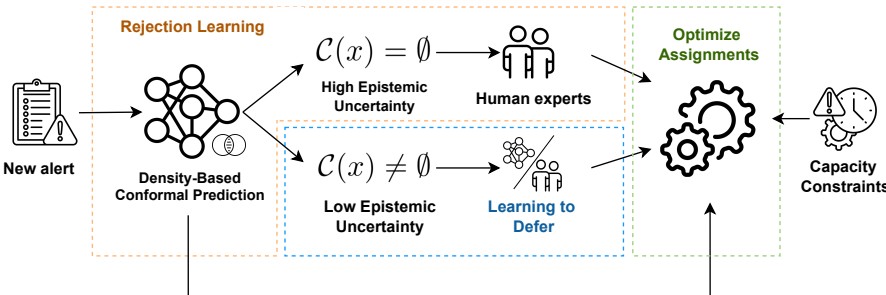

Figure 3: Conformal prediction for HAIC.

**Density-based conformal prediction.** We use density-based conformal prediction (Hechtlinger et al., 2018; Messoudi et al., 2020) as a proxy for epistemic uncertainty by identifying inputs with low support under the training distribution as OOD. Generative density models distinguish high- and low-density regions, with low density commonly associated with OOD and increased epistemic uncertainty (Hüllermeier & Waegeman, 2021; Hechtlinger et al., 2018; Messoudi et al., 2020). This links density-based uncertainty estimation to outlier detection and rejection learning, where models abstain in regions of low support. Importantly, conformal prediction guarantees coverage only under exchangeability, and these guarantees do not extend to arbitrary distribution shift, meaning that conformal prediction is used here as a conservative deferral signal. Limitations are discussed in Section 7.

Density-based conformal prediction uses a two-step training process involving a proper training set $D^{tr} = (X^{tr}, Y^{tr})$ and a calibration set $D^{cal} = (X^{cal}, Y^{cal})$. First, a class-conditional density estimator $\hat{p}(\boldsymbol{x}|y)$ is built using $D^{tr}$. The calibration set is then used to determine the empirical $1 - \alpha$ quantile $\hat{q}_y$ of the density values for each class,

$$\hat{q}_y = \sup\left\{t : \frac{1}{n_y}\sum_{(\boldsymbol{x}_i, y_i) \in D_y^{cal}} \mathbb{I}(\hat{p}(\boldsymbol{x}_i|y) \geq t) \geq 1 - \alpha\right\}, \tag{6}$$

where $\mathbb{I}(\hat{p}(\boldsymbol{x}_i|y) \geq t)$ equals 1 if the estimated probability $\hat{p}(\boldsymbol{x}_i|y)$ is no less than $t$, and 0 otherwise, $D_y^{cal} = \{(\boldsymbol{x}_i, y_i) \in D^{cal} : y_i = y\}$ is the subset of calibration examples in class $y$, and $n_y$ is the cardinality of $D_y^{cal}$. This quantile effectively acts as a threshold, allowing the model to define, for any new observation $x_{n+1}$, a prediction set

$$\mathcal{C}_\alpha(\boldsymbol{x}_{n+1}) = \{y \in \mathcal{Y} : \hat{p}(\boldsymbol{x}_{n+1}|y) \geq \hat{q}_y\}, \tag{7}$$

which includes all classes $y$ for which the observed density is above the threshold. Hechtlinger et al. (2018) show that $|P(y \in \mathcal{C}_\alpha(\boldsymbol{x}_{n+1})) - (1 - \alpha)| \to 0$ as $\min_y n_y \to \infty$, ensuring the asymptotic validity of the model. The training and prediction algorithms for density-based conformal prediction are described in Section B of the Appendix. At inference, the additional computation is limited to evaluating density scores and comparing them to pre-computed thresholds for each class. In binary settings, this adds negligible overhead, but in multi-class scenarios with many labels (e.g., image classification), the cost scales with the number of classes and can become more substantial.

**Aleatoric and epistemic uncertainty.** Figure 3 provides a schematic representation of our system. Instances with high epistemic uncertainty, which deviate significantly from the training distribution, are classified as empty sets and deferred to human experts. For non-empty predictions, which indicate in-distribution instances, the L2D system (as previously formulated in Section 3) assigns instances by using the correctness estimates for both the ML model and human experts. The assignments are then optimized taking into account human work capacity constraints (Section 5).

## 5 Cost and Capacity Constraints

**Cost-sensitive learning.** We adapt our methods to cost-sensitive scenarios in which each instance $i$ incurs a misclassification cost $c_i$. Following Zadrozny et al. (2003), Elkan (2001), and Alves et al. (2024), we replace the standard objective of minimizing the expected error rate $\mathbb{E}_{(\boldsymbol{x},y) \sim D}[\mathbb{I}_{h(\boldsymbol{x}) \neq y}]$, with minimizing the expected misclassification cost $\mathbb{E}_{\boldsymbol{x},y,c \sim D}[c\,\mathbb{I}_{h(\boldsymbol{x}) \neq y}]$, where $D$ is the data distribution and $c$ is the instance-specific cost. To achieve this, Zadrozny et al. (2003) show that we can weight each instance proportionally to its cost by redefining the data distribution as

$$\tilde{D}(\boldsymbol{x}, y) = \frac{c}{\mathbb{E}_{c \sim D}[c]} D(\boldsymbol{x}, y, c), \tag{8}$$

and that, under this modified distribution, the expected error rate is

$$\mathbb{E}_{\boldsymbol{x},y \sim \tilde{D}}[\mathbb{I}_{h(\boldsymbol{x}) \neq y}] = \frac{1}{\mathbb{E}_{c \sim D}[c]} \mathbb{E}_{\boldsymbol{x},y,c \sim D}[c\,\mathbb{I}_{h(\boldsymbol{x}) \neq y}]. \tag{9}$$

Thus, minimizing the error rate under $\tilde{D}$ is equivalent to minimizing the expected misclassification cost under the original distribution $D$. In practical terms, this entails modifying the empirical loss function $\mathcal{L} = \sum_i^N \ell(h(\boldsymbol{x_i}), y_i)$, where $\ell$ is the pointwise loss, by weighing each instance by its associated cost (Zadrozny et al., 2003; Elkan, 2001), resulting in a loss function aligned with the cost-sensitive objective $\mathcal{L} = \sum_{i=1}^N c_i \ell(h(\boldsymbol{x_i}), y_i)$. To implement this approach in our OvA framework, we modify the standard log-loss functions of each of the $|\mathcal{Y}|+J$ binary classifiers. The weighted loss functions are described in detail in Equations 13 and 15 in the Appendix.

**Capacity constraints.** We address human work capacity constraints in both assignment systems. For the density-softmax approach, which is a pure L2D system, we apply the strategy proposed by Alves et al. (2024), optimizing assignments under capacity constraints. For the conformal prediction approach, we extend this framework by creating a method that dynamically balances L2D with ReL while respecting capacity limitations. From a decision-theoretic perspective, this optimization corresponds to maximizing expected utility under uncertainty. Each action consists of routing an instance either to the model or to a specific human expert. The estimated probabilities of correctness serve as instance-wise expected rewards, while misclassification costs and expert capacity constraints define the feasible decision space. The resulting objective can therefore be interpreted as a constrained expected-risk minimization problem, where the system selects the allocation policy that maximizes expected correctness subject to limited human resources.

Rather than processing the entire dataset at once, we impose constraints over batches of instances. This approach aligns with practical scenarios where data is processed in batches and allows the system to adapt

dynamically to varying workload demands. We define a batch vector $\mathbf{b}$ that assigns each instance $i$ to a batch $b$ containing $n_b$ instances, and a human capacity matrix $\mathbf{H}$, where $H_{b,j}$ denotes the maximum number of cases that can be deferred to expert $j$ within batch $b$. Each instance $\boldsymbol{x_i}$ is subject to an action $a_i \in \{1, ..., J+|\mathcal{Y}|\}$ with an associated estimated probability of correctness $\hat{\mathbb{P}}(\text{correct}|\boldsymbol{x_i}, a_i)$, obtained via the OvA classifiers. If $a_i = y + 1$, class $y$ is automatically predicted with an estimated probability of correctness of $\hat{\mathbb{P}}(\text{correct}|\boldsymbol{x_i}, a_i) = \sigma(g_y)$; while $a_i = j + |\mathcal{Y}|$ indicates deferral to the $j$th human expert with an estimated probability of correctness of $\hat{\mathbb{P}}(\text{correct}|\boldsymbol{x_i}, a_i) = \sigma(g_{\perp,j})$. A set of $n_b$ assignments can then be represented as a matrix $A \in \{0,1\}^{n_b \times (J+|\mathcal{Y}|)}$, where $A_{i,a_i}$ denotes whether action $a_i$ is taken for instance $i$. We can then frame the assignment optimization as

$$\boldsymbol{A}^* = \underset{A \in \{0,1\}^{n_b \times (J+|\mathcal{Y}|)}}{\operatorname{argmax}} \sum_{i=1}^{n_b} \sum_{a_i=1}^{J+|\mathcal{Y}|} \hat{\mathbb{P}}(\text{correct}|\boldsymbol{x_i}, a_i) A_{i,a_i}, \tag{10}$$

subject to two constraints: (1) $\sum_{i=1}^{n_b} A_{i,a_i} = H_{b,a_i}$ ensuring each expert meets his capacity, and (2) $\sum_{a_i=1}^{J+|\mathcal{Y}|} A_{i,a_i} = 1$ to ensure each instance has a unique assignment.

For the conformal prediction approach, balancing between L2D and ReL is necessary. The indicator function $\mathbb{I}_{\mathcal{C}_\alpha(\boldsymbol{x_i}) \neq \emptyset}$ indicates whether an instance will be assigned through L2D or ReL, based on the coverage level $\alpha$ for the conformal prediction method. We then optimize this balance alongside the assignment matrix $\boldsymbol{A}$, with the following objective:

$$(\boldsymbol{A}^*, \alpha^*) = \underset{\mathbf{A} \in \{0,1\}^{n_b \times (J+|\mathcal{Y}|)}, \alpha}{\operatorname{argmax}} \sum_{i=1}^{n_b} \sum_{a_i=1}^{J+|\mathcal{Y}|} \left[ \mathbb{I}_{\mathcal{C}_\alpha(\boldsymbol{x_i}) \neq \emptyset} \frac{1}{w_i} \hat{\mathbb{P}}(\text{correct} \mid \boldsymbol{x_i}, a_i) + \mathbb{I}_{\mathcal{C}_\alpha(\boldsymbol{x_i}) = \emptyset} \hat{\mathbb{P}}(\text{correct} \mid X_{\text{train}}, a_i) \right] A_{i,a_i}. \tag{11}$$

This optimization is subject to the same constraints mentioned above, and an additional constraint ensuring that when ReL is applied ($\mathcal{C}_\alpha(\boldsymbol{x_i}) = \emptyset$), the instance is deferred to a human expert, preventing OOD instances from being handled by the model. Additionally, we downweigh model predictions for some instances (through $w_i$) based on the coverage level $\alpha$ at which they fall into the empty set, and if the proportion of empty set predictions exceeds the expected rate from training data. The downweighing is described in detail in Section C. This adaptation allows the system to respond to distribution shifts by prioritizing expert work capacity for OOD instances when required. We solve the assignment problems 10 and 11 using the CP-SAT solver from Google Research's OR-Tools (Perron & Didier).

## 6 Experimental Setup

**Reproducibility.** The code used for the experiments is available at https://github.com/feedzai/Uncertainty-Aware-Systems-for-Human-AI-Collaboration. The dataset is also publicly available (Alves et al., 2025a).

**Dataset.** As discussed in Section 1, datasets with real human predictions are scarce and collecting expert annotations is expensive. OpenL2D (Alves et al., 2025a) addresses this by providing an open-source framework for generating highly customizable synthetic experts with control over feature dependence, bias towards protected attributes, and performance levels. In this work, we use the FiFAR dataset (Alves et al., 2025a), generated with OpenL2D from the publicly available bank-account-fraud (**BAF**) dataset (Jesus et al., 2022). FiFAR simulates a team of fraud analysts whose decision-making properties were validated against the literature and compared to real analysts. In particular, the framework captures instance-dependent expert accuracy, heterogeneity across experts, and realistic inter- and intra-rater variability documented in high-stakes decision-making (Alves et al., 2025a). However, the current setup does not model expert adaptation over time, and while expert behavior is influenced by the model's score, other external factors affecting real-world decisions are not explicitly represented. Our experiments use a team of five analysts, for which the dataset provides predictions on all 30K instances flagged as fraudulent by a fraud detection model, representing an "alert-review" scenario where analysts review high-risk applications. Each instance contains the original **BAF** features, expert predictions, and the fraud model's score. The **BAF** dataset spans 8 months: the first 3 were used in Alves et al. (2025a) to train the fraud detection model, while the flagged 30K instances come from months 4–8 and comprise the FiFAR dataset.

**Misclassification costs.** In account opening fraud detection, a positive prediction leads to application rejection, while a negative prediction results in account opening. Consequently, misclassification costs differ, as false positives (rejecting legitimate applications) lead to customer loss, whereas false negatives (accepting fraudulent applications) may result in financial losses for the bank. Consequently, fraud detection requires balancing the costs of false positives ($c_{FP}$) and false negatives ($c_{FN}$), making commonly used metrics like accuracy inadequate. In our cost-sensitive task, assuming no cost for correct classifications, the objective function to be minimized is the expected misclassification cost,

$$\frac{1}{N} \sum_{i=1}^{N} \left( \lambda \mathbb{I}[y_i = 0 \wedge \hat{y}_i = 1] + \mathbb{I}[y_i = 1 \wedge \hat{y}_i = 0] \right), \tag{12}$$

where $\lambda = c_{FP}/c_{FN}$. As only the ratio $\lambda$ is relevant for minimization purposes, we choose to arbitrarily set $c_{FP} = \lambda$ and $c_{FN} = 1$ for our misclassification cost re-weighting approach. For the FiFAR dataset, $\lambda$ is derived from the alert model threshold selected under the Neyman–Pearson criterion, which maximizes recall at a 5% false positive rate, yielding $\lambda_t = 0.057$ (Alves et al., 2025a).

**Noise injection.** The two foundational works in density-based conformal prediction (Hechtlinger et al., 2018; Messoudi et al., 2020) evaluate robustness by injecting Gaussian noise to simulate distribution shifts. The work introducing density-softmax (Bui & Liu, 2024) similarly applies Gaussian noise among other perturbations in its empirical evaluation. Such noise-based stress tests are standard in uncertainty estimation, where the goal is to assess performance and calibration on borderline or clearly OOD instances (Hein et al., 2019; Hendrycks & Gimpel, 2017; Depeweg et al., 2018). In our case, the introduction of controlled Gaussian noise allows isolating the contributions of each system component, including calibration and performance from density models, and OOD detection and deferral from conformal prediction. We modify the test set (last month of data) by introducing noise to 20% of instances under three configurations: low, medium, and high noise. For each configuration, numerical features are standardized, perturbed with Gaussian noise ($\sigma = 1.0$, 1.5, 2.0), and mapped back to the original scale, with discrete values rounded and bounded features clipped. Categorical and binary features are randomly switched with probabilities 0.3, 0.4, and 0.5, respectively. To account for statistical variation, each configuration is repeated five times with different seeds.

**Data availability and capacity constraints.** We simulate scenarios with variable amounts of expert labels across four data conditions: all instances labeled by all experts, a common but unrealistic assumption in prior L2D methods (Verma et al., 2023; Hemmer et al., 2022), and subsets where experts label only 1/5, 1/20, or 1/40 of the data. These subsets are randomly distributed across experts using five random seeds per condition. In both systems, the expert-specific models are trained on the subset of data with labels for that expert, and in the density-softmax approach, the same subsets train the density models. We impose uniform work capacity constraints on experts by varying deferral rates (10%-50%) to evaluate the system's ability to manage OOD instances across four experimental variables: noise (low, medium, high), number of experts (1-5), deferral rates, and data availability. This leads to 300 different test settings, so results are analyzed for a key subset of representative settings to assess the system's robustness under these constraints.

**Baselines.** We consider three baseline approaches. First, for the **L2D baseline**, we implement the OvA L2D algorithm by Verma et al. (2023) using LightGBM (Ke et al., 2017) models for each binary classification task, omitting the distance-awareness feature to isolate the impact of our proposed contributions. This setup also serves as a baseline for the conformal prediction approach, as the same L2D method handles instances not identified as OOD. Second, in the **ReL baseline**, density-based conformal prediction measures uncertainty to decide whether to defer instances to human experts, deferring randomly when predicted as the empty set and assigning the remainder to the ML model, with coverage level $\alpha$ selected to match expert work capacity. This baseline serves as a measure of the effectiveness of conformal prediction in detecting and deferring OOD instances, independent of any optimization in the expert assignment step. Third, a **random assignment baseline** allocates a random subset of test instances to experts, ensuring capacity constraints are met.

**System Training.** Both systems share a single MLP trained on the data from months 4 to 6 to minimize weighted log-loss, with month 7 used for validation. The MLP's penultimate dense layer is used as a feature extractor to map inputs into a continuous latent space for density estimation. In the density-based conformal prediction method, class-conditional densities are estimated using kernel density estimation

(**KDE**) (Rosenblatt, 1956), and empirical $1 - \alpha$ quantiles are computed on calibration data from month 7. For the density-softmax approach, RealNVP (Dinh et al., 2017) models are trained on the extracted features to estimate likelihoods, using subsets of the data labeled by each expert to reflect specific data availability scenarios. Both systems use LightGBM (Ke et al., 2017) models for the binary classifiers in L2D: the main classifier is trained to predict the true labels, while expert-specific models estimate correctness. Because human experts often have access to the alert model's score, the human correctness models include it as an additional input feature. All classifiers are trained to minimize the weighted log-loss, as described in Section A.1. Details on the training and hyper-parameter selection are available in Section A of the Appendix.

## 7 Results

**Calibration improvements from the density-softmax approach.** To assess how the density model interacts with predictions, we first extract the features from the penultimate layer of the MLP trained for feature extraction and project them in two dimensions with t-SNE. The first panel on the left of Figure 4 shows the resulting feature map in the high-noise test setting, where in-distribution and OOD instances are clearly separated. The second panel displays density scores estimated by the normalizing flows density model, confirming the model's ability to capture the data likelihood under the prior distribution. The last two panels show the effect on classifier predictions: raw LightGBM probabilities (third panel) can be overconfident on OOD regions, but after adjustment with density scores (forth panel), probabilities in those regions are pulled closer to 0.5. This demonstrates how the density-softmax approach improves calibration by adjusting predictions on data unlikely to belong to the training distribution.

Incorporating density models into our L2D classifiers consistently enhances calibration across all settings. Figure 5 reports the mean expected calibration error (**ECE**) with 95% confidence intervals, accounting for the randomness in noise injection and training set selection. Here, data availability is the fraction of training data with a given expert's predictions: 1/1 means full coverage, while 1/40 means each expert labels only 1/40 of the data, and their models (both LightGBM and density) are trained on that subset. The figure shows ECE for the models that predict expert correctness, which are used in our system (Section 3) to assign instances to the most reliable expert. Calibration of these models is crucial, since we directly compare their predicted probabilities of correctness to decide which expert should handle a given instance.

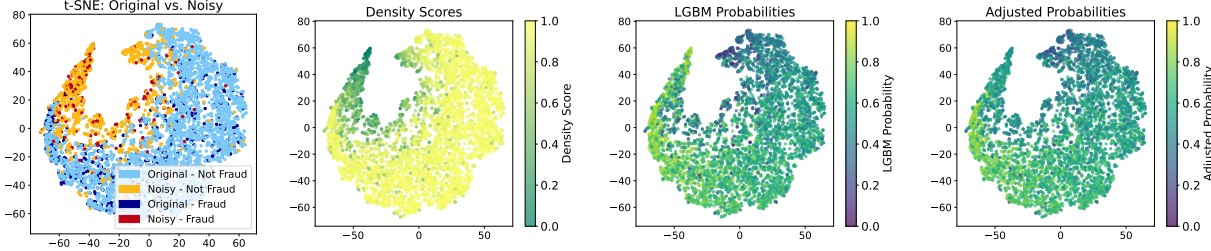

Figure 4: Effect of density-softmax on predictions under high noise. From left to right: (1) t-SNE visualization of extracted feature representations showing separation between in-distribution and OOD instances; (2) density scores estimated by the normalizing flow model, with low-density regions corresponding to OOD data; (3) raw LightGBM predicted probabilities, which are overconfident in OOD regions; (4) adjusted probabilities after density-softmax scaling, where predictions in low-density regions are moderated toward 0.5.

The results show that without density models, ECE increases sharply as noise grows. On the other hand, incorporating density models significantly reduces the growth of ECE, creating a widening gap in performance as noise increases. The benefit is especially pronounced under higher data availability, where having more expert predictions allows for more accurate modeling of human behavior and better estimation of density scores, which in turn improves discrimination between in-distribution and OOD data. This improvement arises because density models balance the expert's predicted probability of correctness with their average correctness on the training data in regions of high epistemic uncertainty (as detailed in Section 3), yielding more reliable predictions even for instances that deviate from the training distribution.

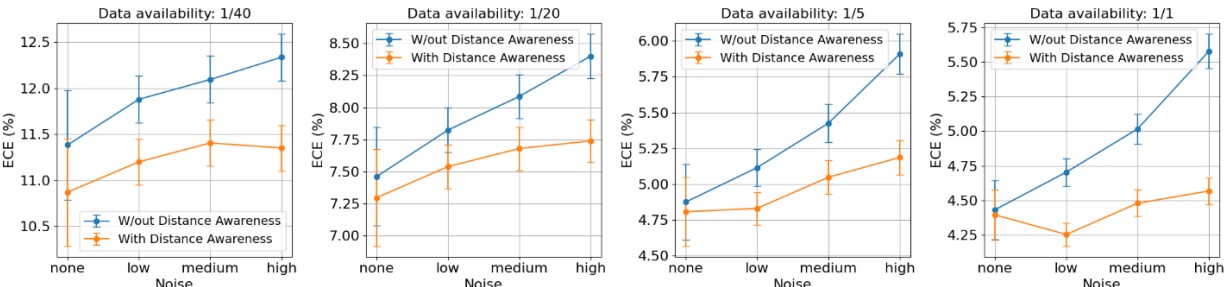

Figure 5: Calibration of expert correctness models with and without density awareness. Mean ECE with 95% confidence intervals across different noise levels and data availability settings. Data availability denotes the fraction of training instances labeled by each expert (1/1 = full coverage; 1/40 = sparse labeling).

**Epistemic uncertainty and calibration in the conformal prediction approach.** In our conformal prediction approach (Section 4), empty-set predictions are used as a proxy for high epistemic uncertainty, controlled by the coverage level $\alpha$: higher $\alpha$ yields more empty sets. These instances are handled via a ReL strategy, while non-empty predictions are assigned through L2D. Ideally, classifiers should exhibit low ECE on non-empty predictions (in-distribution instances, where probability estimates should be reliable) and higher ECE on empty-set predictions (OOD instances, which are deferred to human experts). To examine this, Figure 6 shows the ECE of instances predicted as empty sets across coverage levels $\alpha$. The right plot averages results across expert models with 95% confidence intervals for the mean. ECE is initially high at small $\alpha$ and decreases as $\alpha$ increases across all noise settings. This pattern is consistent with the empirical behavior of density-based conformal prediction: OOD instances tend to receive low density scores and are therefore classified as empty sets, enabling deferral in shifted regimes.

These results highlight the need to choose an optimal coverage level $\alpha$ to balance OOD detection against errors on in-distribution data. In practice, this means optimizing the use of L2D on in-distribution instances and deferring OOD cases to human experts. Consequently, jointly optimizing $\alpha$ and the assignment matrix (Equation 11) is necessary. Results on tuning $\alpha$ are presented in Section D of the Appendix.

Importantly, conformal prediction guarantees coverage only under exchangeability, and its behavior under distribution shift is not theoretically guaranteed. In settings where shift corresponds to movement into low-density regions, this mechanism performs well in practice, as observed here. However, in more complex or multimodal distributions, or under conditional shifts that preserve density while altering label structure, this correspondence may break down and requires empirical validation.

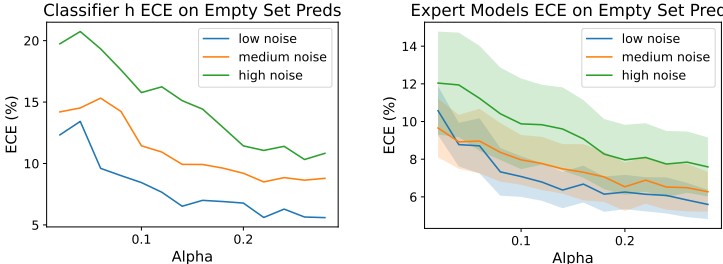

Figure 6: Calibration behavior on instances predicted as empty sets by conformal prediction. Left: classifier ECE on empty-set instances across coverage levels $\alpha$. Right: average ECE of expert correctness models on empty-set instances (shaded regions denote 95% confidence intervals). Smaller $\alpha$ values identify only extreme OOD cases (higher ECE), while increasing $\alpha$ includes more moderately shifted instances, reducing ECE.

**Misclassification cost analysis.** We evaluate the misclassification cost across different deferral strategies and experimental settings. Table 1 compares five strategies (Density-Softmax (**DS**), Conformal Prediction (**CP**), L2D, ReL, and random assignment) showing representative settings from the 300 test cases described

in Section 6, covering three noise levels (low, medium, high), two deferral rates (20% or 40%), 1, 3 or 5 experts, and two data availability scenarios (full data and one-fifth). Each setting is run five times with distinct seeds for data sampling and noise generation, and results are reported with 95% confidence intervals for the mean. Tables with the remaining 300 settings are available at https://anonymous.4open.science/r/Uncertainty-Aware-Systems-for-Human-AI-Collaboration-FB62/results/results.ipynb.

Table 1: Misclassification cost for the representative settings. The best result is in bold and the second best is underlined.

| Setting | | | | Deferral Strategy | | | | |
|---|---|---|---|---|---|---|---|---|
| Noise | NE | DR | DA | CP | DS | L2D | ReL | Random |
| low | 1 | 20 | 1/1 | $\underline{4.81 \pm_{0.11}}$ | $\mathbf{4.81 \pm_{0.17}}$ | $5.09 \pm_{0.13}$ | $5.11 \pm_{0.07}$ | $5.20 \pm_{0.15}$ |
| low | 1 | 20 | 1/5 | $\underline{4.85 \pm_{0.16}}$ | $\mathbf{4.83 \pm_{0.22}}$ | $5.18 \pm_{0.16}$ | $5.11 \pm_{0.07}$ | $5.23 \pm_{0.11}$ |
| low | 1 | 40 | 1/1 | $\mathbf{4.42 \pm_{0.09}}$ | $\underline{4.48 \pm_{0.11}}$ | $5.00 \pm_{0.13}$ | $4.77 \pm_{0.07}$ | $4.97 \pm_{0.13}$ |
| low | 1 | 40 | 1/5 | $\mathbf{4.40 \pm_{0.13}}$ | $\underline{4.41 \pm_{0.09}}$ | $4.89 \pm_{0.15}$ | $4.77 \pm_{0.07}$ | $4.99 \pm_{0.09}$ |
| low | 3 | 20 | 1/1 | $\underline{4.78 \pm_{0.13}}$ | $\mathbf{4.72 \pm_{0.11}}$ | $5.20 \pm_{0.13}$ | $5.14 \pm_{0.10}$ | $5.25 \pm_{0.12}$ |
| low | 3 | 20 | 1/5 | $\underline{4.91 \pm_{0.15}}$ | $\mathbf{4.90 \pm_{0.18}}$ | $5.23 \pm_{0.11}$ | $5.14 \pm_{0.11}$ | $5.28 \pm_{0.18}$ |
| low | 3 | 40 | 1/1 | $\underline{4.40 \pm_{0.07}}$ | $\mathbf{4.35 \pm_{0.09}}$ | $5.11 \pm_{0.13}$ | $4.77 \pm_{0.07}$ | $5.12 \pm_{0.16}$ |
| low | 3 | 40 | 1/5 | $\underline{4.53 \pm_{0.26}}$ | $\mathbf{4.45 \pm_{0.26}}$ | $5.03 \pm_{0.16}$ | $4.76 \pm_{0.08}$ | $5.16 \pm_{0.15}$ |
| low | 5 | 20 | 1/1 | $\underline{4.76 \pm_{0.15}}$ | $\mathbf{4.70 \pm_{0.09}}$ | $5.20 \pm_{0.13}$ | $5.09 \pm_{0.06}$ | $5.30 \pm_{0.21}$ |
| low | 5 | 20 | 1/5 | $\underline{4.85 \pm_{0.10}}$ | $\mathbf{4.84 \pm_{0.16}}$ | $5.19 \pm_{0.14}$ | $5.12 \pm_{0.04}$ | $5.25 \pm_{0.12}$ |
| low | 5 | 40 | 1/1 | $\underline{4.46 \pm_{0.10}}$ | $\mathbf{4.44 \pm_{0.09}}$ | $4.71 \pm_{0.13}$ | $4.79 \pm_{0.09}$ | $5.11 \pm_{0.11}$ |
| low | 5 | 40 | 1/5 | $\underline{4.48 \pm_{0.16}}$ | $\mathbf{4.42 \pm_{0.16}}$ | $4.80 \pm_{0.10}$ | $4.79 \pm_{0.11}$ | $5.07 \pm_{0.18}$ |
| medium | 1 | 20 | 1/1 | $\mathbf{4.82 \pm_{0.07}}$ | $4.96 \pm_{0.09}$ | $5.32 \pm_{0.08}$ | $\underline{4.85 \pm_{0.06}}$ | $5.39 \pm_{0.03}$ |
| medium | 1 | 20 | 1/5 | $\mathbf{4.82 \pm_{0.09}}$ | $4.97 \pm_{0.15}$ | $5.37 \pm_{0.06}$ | $\underline{4.85 \pm_{0.06}}$ | $5.46 \pm_{0.15}$ |
| medium | 1 | 40 | 1/1 | $\mathbf{4.41 \pm_{0.05}}$ | $4.61 \pm_{0.10}$ | $5.20 \pm_{0.08}$ | $\underline{4.49 \pm_{0.09}}$ | $5.13 \pm_{0.09}$ |
| medium | 1 | 40 | 1/5 | $\mathbf{4.46 \pm_{0.12}}$ | $4.60 \pm_{0.12}$ | $5.08 \pm_{0.11}$ | $\underline{4.49 \pm_{0.09}}$ | $5.17 \pm_{0.24}$ |
| medium | 3 | 20 | 1/1 | $\mathbf{4.85 \pm_{0.09}}$ | $4.91 \pm_{0.09}$ | $5.42 \pm_{0.08}$ | $\underline{5.01 \pm_{0.13}}$ | $5.47 \pm_{0.12}$ |
| medium | 3 | 20 | 1/5 | $\underline{5.00 \pm_{0.11}}$ | $5.08 \pm_{0.19}$ | $5.39 \pm_{0.08}$ | $\mathbf{4.99 \pm_{0.14}}$ | $5.50 \pm_{0.14}$ |
| medium | 3 | 40 | 1/1 | $\mathbf{4.41 \pm_{0.08}}$ | $\underline{4.52 \pm_{0.06}}$ | $5.29 \pm_{0.08}$ | $4.55 \pm_{0.11}$ | $5.39 \pm_{0.13}$ |
| medium | 3 | 40 | 1/5 | $\underline{4.56 \pm_{0.19}}$ | $4.70 \pm_{0.28}$ | $5.20 \pm_{0.15}$ | $\mathbf{4.50 \pm_{0.09}}$ | $5.40 \pm_{0.12}$ |
| medium | 5 | 20 | 1/1 | $\underline{4.86 \pm_{0.07}}$ | $4.94 \pm_{0.13}$ | $5.41 \pm_{0.08}$ | $\mathbf{4.86 \pm_{0.05}}$ | $5.51 \pm_{0.08}$ |
| medium | 5 | 20 | 1/5 | $\mathbf{4.86 \pm_{0.12}}$ | $5.00 \pm_{0.09}$ | $5.36 \pm_{0.14}$ | $\underline{4.87 \pm_{0.07}}$ | $5.47 \pm_{0.11}$ |
| medium | 5 | 40 | 1/1 | $\mathbf{4.37 \pm_{0.08}}$ | $\underline{4.51 \pm_{0.07}}$ | $5.03 \pm_{0.08}$ | $4.54 \pm_{0.11}$ | $5.35 \pm_{0.14}$ |
| medium | 5 | 40 | 1/5 | $\mathbf{4.47 \pm_{0.12}}$ | $4.60 \pm_{0.16}$ | $5.06 \pm_{0.16}$ | $\underline{4.53 \pm_{0.12}}$ | $5.27 \pm_{0.13}$ |
| high | 1 | 20 | 1/1 | $\mathbf{4.62 \pm_{0.07}}$ | $5.00 \pm_{0.09}$ | $5.44 \pm_{0.13}$ | $\underline{4.86 \pm_{0.07}}$ | $5.50 \pm_{0.14}$ |
| high | 1 | 20 | 1/5 | $\mathbf{4.65 \pm_{0.14}}$ | $4.97 \pm_{0.19}$ | $5.49 \pm_{0.13}$ | $\underline{4.86 \pm_{0.07}}$ | $5.48 \pm_{0.10}$ |
| high | 1 | 40 | 1/1 | $\mathbf{4.21 \pm_{0.08}}$ | $4.53 \pm_{0.08}$ | $5.23 \pm_{0.13}$ | $\underline{4.39 \pm_{0.04}}$ | $5.25 \pm_{0.22}$ |
| high | 1 | 40 | 1/5 | $\mathbf{4.22 \pm_{0.10}}$ | $4.56 \pm_{0.13}$ | $5.15 \pm_{0.15}$ | $\underline{4.39 \pm_{0.04}}$ | $5.25 \pm_{0.17}$ |
| high | 3 | 20 | 1/1 | $\mathbf{4.74 \pm_{0.13}}$ | $\underline{5.04 \pm_{0.15}}$ | $5.50 \pm_{0.13}$ | $5.09 \pm_{0.19}$ | $5.55 \pm_{0.16}$ |
| high | 3 | 20 | 1/5 | $\mathbf{4.81 \pm_{0.17}}$ | $5.15 \pm_{0.16}$ | $5.49 \pm_{0.16}$ | $\underline{5.09 \pm_{0.15}}$ | $5.56 \pm_{0.15}$ |
| high | 3 | 40 | 1/1 | $\mathbf{4.29 \pm_{0.09}}$ | $4.53 \pm_{0.09}$ | $5.41 \pm_{0.13}$ | $\underline{4.46 \pm_{0.09}}$ | $5.40 \pm_{0.17}$ |
| high | 3 | 40 | 1/5 | $\mathbf{4.29 \pm_{0.14}}$ | $4.65 \pm_{0.28}$ | $5.32 \pm_{0.23}$ | $\underline{4.47 \pm_{0.06}}$ | $5.42 \pm_{0.09}$ |
| high | 5 | 20 | 1/1 | $\mathbf{4.72 \pm_{0.15}}$ | $5.04 \pm_{0.13}$ | $5.54 \pm_{0.13}$ | $\underline{4.85 \pm_{0.09}}$ | $5.61 \pm_{0.15}$ |
| high | 5 | 20 | 1/5 | $\mathbf{4.75 \pm_{0.17}}$ | $5.04 \pm_{0.17}$ | $5.49 \pm_{0.11}$ | $\underline{4.86 \pm_{0.10}}$ | $5.59 \pm_{0.17}$ |
| high | 5 | 40 | 1/1 | $\mathbf{4.25 \pm_{0.10}}$ | $4.57 \pm_{0.14}$ | $5.01 \pm_{0.13}$ | $\underline{4.42 \pm_{0.07}}$ | $5.37 \pm_{0.19}$ |
| high | 5 | 40 | 1/5 | $\mathbf{4.25 \pm_{0.10}}$ | $4.56 \pm_{0.13}$ | $5.10 \pm_{0.17}$ | $\underline{4.44 \pm_{0.06}}$ | $5.41 \pm_{0.17}$ |

Across all noise levels, both DS and CP outperform the baselines on average. In low-noise scenarios, DS consistently achieves the lowest misclassification cost, benefiting from its gradual adjustment of probability estimates: by slightly adjusting overconfident predictions in uncertain regions, DS produces more accurate estimates of expert correctness and thus better assignment decisions. Under medium and high noise, CP becomes the strongest strategy. Its hard cutoff mechanism—deferring all instances deemed OOD to experts—provides a clear advantage when noisy data is easier to identify, while DS may still allow some OOD instances to be handled by the classifier. The most informative baseline is ReL. Both ReL and CP use density-based conformal prediction to flag empty-set (OOD) cases, but differ in how the coverage level $\alpha$ is set. ReL fixes $\alpha$ to match expert capacity, rejecting as many instances as possible up to that limit. CP instead optimizes $\alpha$ to balance assignments between L2D and ReL: more OOD data pushes assignments toward ReL, while more in-distribution data favors L2D. This optimization enables CP to interpolate between the two strategies depending on the data distribution, and the CP–ReL performance gap therefore quantifies the added value of assignment and coverage optimization. On the other hand, standard L2D lacks

density information to assign instances correctly, highlighting the importance of having calibrated estimates of correctness in L2D. The DS vs. L2D comparison shows that incorporating density scores improves expert assignment in the presence of OOD data.

**Sensitivity and robustness.** Although the two systems rely on different mechanisms, they share the same structural principle: combining an uncertainty signal with deferral policies. Their consistent improvement over L2D and ReL suggests that performance gains stem from uncertainty-guided policy selection rather than any single implementation detail. Regarding hyperparameters, the conformal approach is primarily sensitive to the coverage level $\alpha$, which controls the balance between routing strategies. However, across roughly 300 experimental settings, the optimization procedure consistently identifies appropriate values (see Appendix D). For density-softmax, there are no dedicated hyperparameters beyond those of the underlying ML models. The only additional design choice is the weighting between predicted and average expert correctness, but this varies naturally across settings since each condition assumes different labeled data subsets per expert. Performance therefore does not hinge on a fixed weighting configuration.

**Computational considerations.** Both approaches introduce moderate additional training cost due to density estimation, while maintaining lightweight test-time overhead. In the uncertainty-aware L2D system, density-softmax augments each OvA correctness model with a density estimator, requiring one additional density model per expert (trained on the subset of data labeled by that expert). At inference time, the added cost is limited to computing a density score and applying a simple logit scaling, and overall complexity scales linearly with the number of experts, as in standard OvA L2D. In the conformal prediction approach, the additional component is density-based conformal prediction, which requires training class-conditional density models and computing empirical quantiles on a calibration set. In multiclass settings, the cost scales linearly with the number of classes, as each class requires its own density estimator and threshold. At inference, computation consists of evaluating density scores and comparing them to class-specific thresholds.

## 8 Conclusions and Future Work

We showed that in dynamic environments, human–AI collaboration (HAIC) systems must adapt their routing policy based on epistemic uncertainty, rather than rely on a single deferral strategy. We introduced two uncertainty-aware approaches that outperform traditional learning-to-defer (L2D) and rejection learning (ReL) methods. Both approaches incorporate uncertainty estimation to improve robustness under distribution shift while accounting for cost sensitivity, limited expert data, and human capacity constraints. Empirical results demonstrate that uncertainty-driven routing enables the system to select, for each instance, the appropriate deferral policy according to its estimated epistemic uncertainty.

Although our experiments focus on fraud detection, the proposed framework is domain-agnostic in the sense that it combines an uncertainty signal, deferral policies, and capacity-constrained assignment. These components are generalizable in other high-stakes settings such as healthcare triage and risk assessment. Adapting the approach to a new domain mainly requires selecting an appropriate uncertainty proxy for the data modality, and defining domain-specific costs and capacity constraints. We scope our empirical claims to the tabular alert-review setting, and validating performance in other domains remains future work.

Beyond this, several extensions merit further investigation. Incorporating dynamic expert feedback, for example by retraining on newly labeled cases to account for evolving expert strengths, would better reflect real-world human–AI collaboration. Moreover, uncertainty-driven deferral policies directly affect workload distribution and responsibility allocation. If low-density regions correlate with specific subpopulations, systematic deferral may concentrate human scrutiny or delays on those groups. Future research should therefore analyze deferral patterns and explore fairness-aware constraints in deployed HAIC systems.

## Acknowledgments

M. Figueiredo was partially supported by FCT (*Fundação para a Ciência e a Tecnologia*), under project UID/50008/2025–Instituto de Telecomunicações, with DOI `https://doi.org/10.54499/UID/50008/2025`.

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

# A    Training details

## A.1    Learning to Defer

For the ML classifier $h$, we employ a LightGBM classifier (Ke et al., 2017), trained on the alerts raised from months four to six and validated on month 7. We perform hyperparameter optimization using TPE (Akiba et al., 2019), running the optimization process for 300 trials with 200 start-up trials, using the hyperparameter search space in Table 2. As described in Section 5, the goal of this search is to minimize the weighted log-loss

$$\mathcal{L}'_h \left( \{x_i, y_i, c_i\}_{i=1}^N, g \right) = \frac{1}{N} \sum_{i=1}^N c_i \left[ -y_i \log \left( \psi^{-1}\big(g(x_i)\big) \right) - (1 - y_i) \log \left( 1 - \psi^{-1}\big(g(x_i)\big) \right) \right], \quad (13)$$

where $g$ is the scoring function we are training to predict $y_i$, $\psi^{-1}$ is a well-defined inverse link function, and $c_i$ is the instance-specific cost. Note that although in our experiments $c_i$ can only have two possible values ($c_{FP}$ or $c_{FN}$), this formulation can be applied to any misclassification cost structure.

Since false negatives carry a higher cost, we explore whether introducing a bias towards predicting fraud is beneficial for our model. This is done by testing different initial probability estimates for the base predictor in the boosting model. During model training, we perform an independent hyperparameter search over a range of values for the initial prediction value $g_{initial}$, from $g_d$ to $g_d + 2$, in increments of 0.2, where $g_d$ is the default initial prediction value

$$g_d = logit \left( \frac{\sum_i c_i y_i}{\sum_i c_i} \right). \quad (14)$$

In the first hyperparameter search, LightGBM classifiers with a maximum tree depth of 2 achieved the best performance. Therefore, we conducted a second search consisting of a total of 1700 trials, with 1500 startup trials, where we used the same hyperparameter search space detailed in Table 2, but fixing the maximum depth parameter at 2.

Table 2: ML classifier: lightGBM hyperparameter search space.

| Hyperparameter | Values | Distribution |
|---|---|---|
| boosting_type | "dart" | |
| enable_bundle | [False, True] | |
| n_estimators | [50, 250] | Uniform |
| max_depth | [2, 5] | Uniform |
| num_leaves | [100, 1000] | Uniform |
| min_child_samples | [5, 200] | Uniform |
| learning_rate | [0.001, 1] | Uniform |
| reg_alpha | [0.0001, 2] | Uniform |
| reg_lambda | [0.0001, 2] | Uniform |

For the expert correctness models, we also use LightGBM classifiers (Ke et al., 2017). These models are trained on alerts raised during months four to six and validated on alerts from month seven. The objective is to predict the correctness of the experts' predictions, using only data for which human labels are available. Consequently, the amount of training data depends on the data availability scenario under consideration. Hyperparameter optimization is performed using TPE (Akiba et al., 2019), with a total of 120 trials, including 100 start-up trials. The hyperparameter search space is detailed in Table 3. The objective is to minimize the weighted log-loss, adapted to predict expert correctness instead of fraud labels. The loss function is defined as

$$\mathcal{L}'_j\left(\{S_i\}_{i=1}^N, g_{\perp,j}\right) = \frac{1}{N}\sum_{i=1}^N c_i\left[-\mathbb{I}[m_{i,j}=y]\log\left(\psi^{-1}\left(g_{\perp,j}(x_i)\right)\right) - \mathbb{I}[m_{i,j}\neq y]\log\left(1-\psi^{-1}\left(g_{\perp,j}(x_i)\right)\right)\right],$$

(15)

where $S_i = \{x_i, y_i, c_i, m_{i,j}\}$, and $m_{i,j}$ represents a prediction from expert $j$ on instance $x_i$.

Table 3: Expert models: lightGBM hyperparameter search space.

| Hyperparameter | Values | Distribution |
|---|---|---|
| boosting_type | "dart" | |
| enable_bundle | [False, True] | |
| n_estimators | [50, 250] | Uniform |
| max_depth | [2, 20] | Uniform |
| num_leaves | [100, 1000] | Log |
| min_child_samples | [5, 100] | Log |
| learning_rate | [0.005, 0.5] | Log |
| reg_alpha | [0.0001, 0.1] | Log |
| reg_lambda | [0.0001, 0.1] | Log |

## A.2 Density Estimation

Both our systems incorporate a density estimation step. For this, we train an MLP and use its penultimate dense layer as a feature extractor to map inputs into a continuous latent space. The MLP is trained on alerts from months 4 to 6 and validated on month 7. It is optimized to minimize the weighted log-loss described in Equation 13. Hyperparameter optimization is performed using TPE (Akiba et al., 2019), running for 100 trials with 10 startup trials. The hyperparameter search space and the selected parameters are shown in Table 4.

In our density-based conformal prediction method, we apply KDE (Rosenblatt, 1956) with a bandwidth of 0.2 to the feature vectors extracted from the MLP. These density scores are subsequently used to compute prediction sets, as outlined in Section B of the Appendix.

Table 4: Feature extraction MLP: hyperparameter search space.

| Hyperparameter | Values | Distribution | Selected |
|---|---|---|---|
| optimizer | Adam | – | Adam |
| hidden_layer_sizes | {[128/200, 20/50/128, 0/20/50]} | – | [128, 50] |
| learning_rate | [1e-5, 1e-3] | Log | 3.096e-5 |
| weight_decay | [1e-5, 1e-3] | Log | 3.84e-4 |
| dropout_rate | [0.0, 0.5] | Uniform | 0.056 |
| num_epochs | [20, 200] | Uniform | 73 |
| batch_size | [10, 128] | Log | 30 |

In the density-softmax approach, we use RealNVP normalizing flows (Dinh et al., 2017) to estimate data likelihoods from the extracted features. Hyperparameter optimization for RealNVP is performed using TPE (Akiba et al., 2019) over 100 trials (10 startup trials) to minimize the negative log-likelihood on validation data. Training data spans alerts from months 4 to 6, with validation on month 7. The search space for hyperparameters is detailed in Table 5. A separate RealNVP model is then trained for each expert and data availability scenario, producing unique density scores for each expert based on their labeled subsets.

Table 5: RealNVP: hyperparameter search space.

| Hyperparameter | Values | Distribution |
|---|---|---|
| num_coupling_layers | [2, 10] | Uniform |
| hidden_dim | [64, 256] | Uniform (step 32) |
| learning_rate | [1e-5, 1e-3] | Log |
| num_epochs | [50, 300] | Uniform |

## B  Density-Based Conformal Prediction

The training and prediction algorithms for density-based conformal prediction are defined in Algorithms 1 and 2.

---
**Algorithm 1:** Training algorithm

---
**Input:** Training data $Z = (x_i, y_i), i = 1 \ldots n$, Class list $\mathcal{Y}$, Confidence level $\alpha$, Ratio $p$
**Output:** $\hat{p}_{\text{list}}, \hat{q}_{\text{list}}$
**Initialize:** $\hat{p}_{\text{list}} = \text{list}, \hat{q}_{\text{list}} = \text{list}$
**for** $y \in \mathcal{Y}$ **do**
  $X_y^{tr}, X_y^{cal} \leftarrow \text{SubsetData}(Z, \mathcal{Y}, p)$
  $\hat{p}_y \leftarrow \text{LearnDensityEstimator}(X_y^{tr})$
  $\hat{q}_y \leftarrow \text{Quantile}(\hat{p}_y(X_y^{cal}), \alpha)$
  $\hat{p}_{\text{list}}.\text{append}(\hat{p}_y)$
  $\hat{q}_{\text{list}}.\text{append}(\hat{q}_y)$
**end for**
**return** $\hat{p}_{list}, \hat{q}_{list}$

---

---

**Algorithm 2:** Prediction algorithm

---

**Input:** Input $x$, Trained $\hat{p}_{\text{list}}, \hat{q}_{\text{list}}$, Class list $\mathcal{Y}$
**Output:** $\mathcal{C}$
**Initialize:** $\mathcal{C} = $ list
**for** $y \in \mathcal{Y}$ **do**
    **if** $\hat{p}_y(x) \geq \hat{q}_y$ **then**
        $\mathcal{C}$.append($y$)
    **end if**
**end for**
**return** $\mathcal{C}$

---

## C   Downweighing model prediction in the conformal prediction approach

The downweighting factor $\frac{1}{w_i}$ in Equation 11 is determined by the conformal coverage level $\alpha$ and the proportion of empty set predictions in the batch. For each new instance, we compute two density scores $\hat{p}(x_i|y)$ (one for each class in a binary setting). If neither score exceeds the quantile $\hat{q}_y$ defined during calibration, the instance is classified into the empty set. These quantiles are selected to meet the desired coverage level $\alpha$. Lower $\alpha$ values represent a more conservative approach to detecting OOD data, prioritizing accuracy and reducing empty set predictions, while higher $\alpha$ allows for more empty set predictions with a higher error rate (Figure 7). For each instance, we can find the smallest alpha $\alpha_\emptyset$ for which the instance is predicted as the empty set $\alpha$ by gradually increasing the coverage level $\alpha$. This process requires minimal additional computation since the density scores are already available; we simply need to find the $1 - \alpha$ quantile. The smaller $\alpha_\emptyset$, the more OOD the instance is considered, and one approach is to set $w_i = \alpha_\emptyset$.

It's important to differentiate between instances classified as empty sets due to the desired coverage level and those classified as empty sets due to data drift. In the absence of data drift, the proportion of empty set predictions should approximate $\alpha$ but should not exceed it. If the proportion of empty set predictions does exceed $\alpha$, this suggests that accuracy has fallen below $1 - \alpha$, indicating a lack of data exchangeability and confirming the presence of drift (as illustrated in Figure 7). Consequently, we should only downweigh instances classified as empty sets due to data drift, rather than those classified as empty sets to meet the coverage level.

To build on this reasoning, we can detect OOD data by monitoring the ratio $\rho_{\emptyset,\alpha_\emptyset}/\alpha_\emptyset$, where $\rho_{\emptyset,\alpha_\emptyset}$ represents the proportion of empty set predictions at a specific coverage level $\alpha_\emptyset$. This ratio compares the actual proportion of empty set predictions with the expected proportion based on $\alpha_\emptyset$. A high ratio suggests that the proportion of empty set predictions is higher than what is anticipated under normal circumstances, indicating potential data drift. By calculating this ratio for each instance, we can distinguish empty set predictions due to OOD data from those due to the desired coverage level $\alpha_\emptyset$.

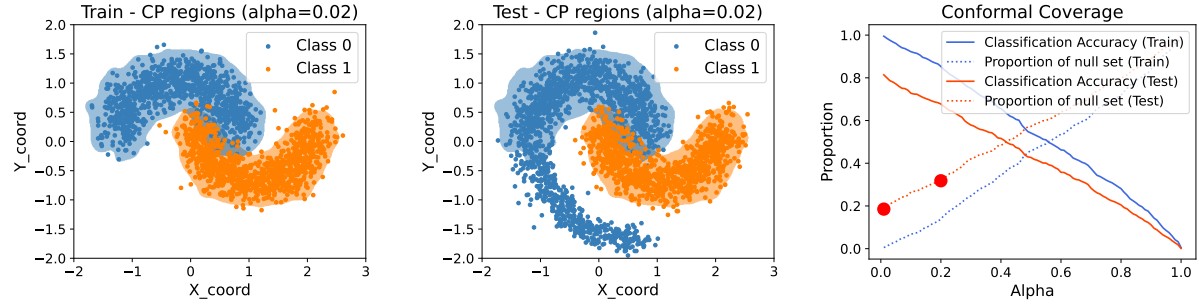

Figure 7: Example of how density-based conformal prediction is affected by data drift. At test time the distribution changes significantly, increasing the number of empty set predictions.

For example, in Figure 7, consider instances predicted as empty sets at $\alpha_\emptyset = 0$. Since $\rho_{\emptyset,\alpha_\emptyset} = 0.2$, the ratio becomes $\frac{0.2}{0} = \infty$, leading to these instances being fully downweighted with $\frac{1}{\infty} = 0$. For instances with $\alpha_\emptyset = 0.2$, the ratio is $\frac{0.4}{0.2} = 2$, resulting in a downweighing factor of $\frac{1}{2}$.

One challenge here is that an instance predicted as an empty set at a particular $\alpha_\emptyset$ will also remain an empty set for all higher $\alpha$ values. This inflates the empty set proportion $\rho_{\emptyset,\alpha_\emptyset}$, not specifically due to the amount of empty sets at $\alpha_\emptyset$, but due to previously identified empty sets at lower $\alpha$ levels. For instance, in the previous example, all OOD data is detected for $\alpha$ as low as 0.02; however, the instance that is predicted as the empty set for $\alpha = 0.2$ is still downweighed by 0.5.

To address this, rather than directly comparing $\rho_{\emptyset,\alpha_\emptyset}$ with $\alpha_\emptyset$, we fit a line between the points $(\alpha_\emptyset - s, \rho_{\emptyset,\alpha_\emptyset - s})$ and $(1, 1)$, where $s$ is the step size between successive values of $\alpha$. This approach is based on the assumption that at $\alpha = 1$, we expect 100% of instances to be classified as empty sets. The value of this line provides a heuristic for the expected proportion of empty sets, assuming no additional OOD data is identified as $\alpha$ increases. Therefore, this adjustment allows determining if additional OOD data is identified at $\alpha_\emptyset$. The resulting weight $w_i$ for an instance $x_i$ is then calculated as:

$$w(x_i, \mathcal{X}_{\text{batch}}, \mathcal{C}_{\{\alpha_i\}_{i=1}^n}) = \frac{\rho_{\emptyset,\alpha_\emptyset}}{\rho_{\emptyset,\alpha_\emptyset - s} + \frac{1 - \rho_{\emptyset,\alpha_\emptyset - s}}{1 - (\alpha_\emptyset - s)} s}, \tag{16}$$

where $\rho_{\emptyset,\alpha_\emptyset} = \frac{|\mathcal{C}_{\alpha_\emptyset}(\mathcal{X}_{\text{batch}}) = \emptyset|}{|\mathcal{X}_{\text{batch}}|}$ represents the proportion of empty set predictions at $\alpha_\emptyset$, based on the instance $x_i$ and the batch $\mathcal{X}_{\text{batch}}$. The denominator provides the expected proportion of empty sets, derived from fitting the line as described, where $s$ is the step size between successive values of $\alpha$. If $\alpha_\emptyset = 0$, the denominator is set to 0, ensuring that fully OOD instances are appropriately downweighted.

By applying this approach to the example in Figure 7, the first point in the conformal coverage plot (corresponding to $\alpha_\emptyset = 0$) would still be significantly downweighed, as the high proportion of empty set predictions indicates a clear detection of OOD data. However, the second point, which corresponds to a higher $\alpha_\emptyset$, would not be downweighed at all, since all OOD data had already been identified at lower $\alpha$ levels. This behavior is exactly what we want, as it prevents the system from penalizing in-distribution data that comes after OOD data has been detected at lower $\alpha_\emptyset$.

## D   Optimizing the conformal coverage level

In our conformal prediction framework, an empty set prediction indicates high epistemic uncertainty, controlled by the coverage level $\alpha$: a higher $\alpha$ leads to more empty set predictions overall. Figure 8 illustrates two complementary views of this behavior. The left plot shows, for noisy (OOD) instances only, the proportion predicted as the empty set as $\alpha$ increases. The right plot shows the proportion of empty set predictions corresponding to noisy instances. Note that when $\alpha = 1$, all instances are predicted as the empty set, which explains why, in the right plot, the $y$-axis lower bound is 0.2 (the fraction of noisy data we introduced) and why in the left plot all noisy data is eventually identified.

The key question is how easily (i.e., at low $\alpha$ values, where only a few empty sets are allowed overall) noisy data can be detected. In the high-noise setting, even for very low $\alpha$, nearly all noisy instances are already identified as empty sets, with the right plot showing that these empty set predictions almost exclusively originate from noisy data. By contrast, in the low-noise setting, noisy data is harder to separate: the proportion of noisy instances predicted as empty sets is lower, with a substantial fraction of empty set predictions coming from in-distribution data. This is expected, as in this setting the noisy data is more similar to the original distribution. Nevertheless, even under low noise, at very small $\alpha$ values, roughly 60% of empty set predictions still correspond to noisy data, meaning the method identifies the most evident OOD instances. These results confirm the method's ability to identify noisy instances and defer such cases via ReL, especially in high noise scenarios.

In Table 6, we present the proportion of empty set predictions (with 95% confidence intervals) for each experimental setting. These results are averaged over five runs, each using a different random seed for noise injection.

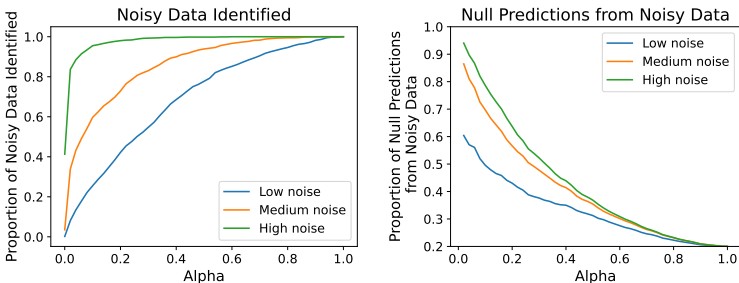

Figure 8: Conformal prediction for OOD detection. Left: proportion of noisy (OOD) instances predicted as empty sets. Right: proportion of empty set predictions originating from noisy data.

Our optimization process (Equation 11) is highly dependent on both the noise setting and the deferral rate. The noise setting controls the degree of distributional shift in the test set: higher noise levels lead to more empty set predictions at lower coverage levels. For instance, in the low-noise scenario, fewer than 4% of instances on average are predicted as the empty set; in contrast, under high noise, this proportion can rise to 20%, reflecting the fact that around 20% of the test instances are corrupted.

The deferral rate further influences the optimization process. As the deferral rate increases, the solver selects a coverage level $\alpha$ that typically yields a higher fraction of empty sets, ensuring that the number of cases referred to human experts remains within their workload constraints. Consistently across all settings, we observe a rising percentage of empty set predictions with increasing deferral rates.

Table 6: Conformal Coverage Optimization for the representative settings.

| Setting | | | | Empty set predictions | |
|---|---|---|---|---|---|
| Noise | NE | DR | DA | ∅ % | Alpha |
| low | 1 | 20 | 1/1 | 4.29% ± 0.66 | 0.03 ± 0.0068 |
| low | 1 | 20 | 1/5 | 3.08% ± 0.67 | 0.02 ± 0.0054 |
| low | 1 | 40 | 1/1 | 8.23% ± 0.63 | 0.07 ± 0.0064 |
| low | 1 | 40 | 1/5 | 4.61% ± 0.88 | 0.03 ± 0.0090 |
| low | 3 | 20 | 1/1 | 2.57% ± 0.32 | 0.01 ± 0.0026 |
| low | 3 | 20 | 1/5 | 2.38% ± 0.42 | 0.01 ± 0.0024 |
| low | 3 | 40 | 1/1 | 4.11% ± 0.44 | 0.03 ± 0.0055 |
| low | 3 | 40 | 1/5 | 3.55% ± 0.67 | 0.02 ± 0.0059 |
| low | 5 | 20 | 1/1 | 2.58% ± 0.23 | 0.01 ± 0.0018 |
| low | 5 | 20 | 1/5 | 2.37% ± 0.45 | 0.01 ± 0.0032 |
| low | 5 | 40 | 1/1 | 3.65% ± 0.59 | 0.02 ± 0.0054 |
| low | 5 | 40 | 1/5 | 3.25% ± 0.51 | 0.02 ± 0.0037 |
| medium | 1 | 20 | 1/1 | 9.93% ± 0.42 | 0.03 ± 0.0034 |
| medium | 1 | 20 | 1/5 | 9.03% ± 0.44 | 0.03 ± 0.0022 |
| medium | 1 | 40 | 1/1 | 15.04% ± 0.58 | 0.08 ± 0.0067 |
| medium | 1 | 40 | 1/5 | 10.94% ± 1.14 | 0.04 ± 0.0085 |
| medium | 3 | 20 | 1/1 | 8.47% ± 0.13 | 0.02 ± 0.0013 |
| medium | 3 | 20 | 1/5 | 8.35% ± 0.25 | 0.02 ± 0.0023 |
| medium | 3 | 40 | 1/1 | 9.77% ± 0.43 | 0.03 ± 0.0015 |
| medium | 3 | 40 | 1/5 | 9.56% ± 0.53 | 0.03 ± 0.0019 |
| medium | 5 | 20 | 1/1 | 8.55% ± 0.06 | 0.02 ± 0.0014 |
| medium | 5 | 20 | 1/5 | 8.28% ± 0.11 | 0.02 ± 0.0019 |
| medium | 5 | 40 | 1/1 | 9.69% ± 0.51 | 0.03 ± 0.0022 |
| medium | 5 | 40 | 1/5 | 9.44% ± 0.43 | 0.03 ± 0.0027 |
| high | 1 | 20 | 1/1 | 17.03% ± 0.59 | 0.02 ± 0.0020 |
| high | 1 | 20 | 1/5 | 16.62% ± 0.95 | 0.02 ± 0.0023 |
| high | 1 | 40 | 1/1 | 20.19% ± 0.61 | 0.05 ± 0.0073 |
| high | 1 | 40 | 1/5 | 18.33% ± 0.29 | 0.03 ± 0.0034 |
| high | 3 | 20 | 1/1 | 15.12% ± 0.88 | 0.01 ± 0.0023 |
| high | 3 | 20 | 1/5 | 14.93% ± 1.00 | 0.01 ± 0.0025 |
| high | 3 | 40 | 1/1 | 17.90% ± 0.17 | 0.02 ± 0.0012 |
| high | 3 | 40 | 1/5 | 17.75% ± 0.25 | 0.02 ± 0.0008 |
| high | 5 | 20 | 1/1 | 16.25% ± 0.95 | 0.02 ± 0.0028 |
| high | 5 | 20 | 1/5 | 15.96% ± 0.88 | 0.02 ± 0.0022 |
| high | 5 | 40 | 1/1 | 17.79% ± 0.19 | 0.02 ± 0.0009 |
| high | 5 | 40 | 1/5 | 17.72% ± 0.24 | 0.02 ± 0.0020 |

