# OpenReview forum: "Uncertainty-Aware Systems for Human-AI Collaboration"
_TMLR — Accepted by TMLR_

### Review · Reviewer_5ow4 · 2025-12-22

**Summary Of Contributions:**

This paper addresses a central limitation of learning-to-defer (L2D) systems for human–AI collaboration: their brittleness under distribution shift due to unreliable confidence and correctness estimates. The authors propose two uncertainty-aware HAIC frameworks that explicitly incorporate epistemic uncertainty into the decision-allocation process.

First, the paper introduces an uncertainty-aware L2D system that augments the OvA L2D framework with density-softmax models based on normalizing flows. By adjusting classifier and expert-correctness logits using density estimates in latent space, the method reduces overconfidence on out-of-distribution (OOD) instances and improves calibration. The approach further blends instance-level predictions with experts’ historical average performance under high epistemic uncertainty, enabling more robust expert assignment when data is scarce or shifted.

Second, the paper proposes a hybrid conformal prediction–based HAIC system that dynamically switches between L2D and rejection learning (ReL). Density-based conformal prediction is used to identify OOD instances (empty prediction sets), which are directly deferred to human experts, while in-distribution instances are handled via L2D. The coverage level is optimized jointly with expert assignment under cost-sensitive objectives and human capacity constraints.

**Audience:**

Yes

**Audience Explanation:**

The paper would be of clear interest to segments of TMLR’s audience working on learning to defer, uncertainty estimation, human–AI collaboration, selective prediction, and decision-making under distribution shift. It addresses a well-known limitation of existing L2D methods and proposes principled, uncertainty-aware extensions with practical relevance, including cost sensitivity and capacity constraints. The combination of methodological contributions and a thorough empirical study makes the findings valuable to researchers interested in robust ML systems deployed in real-world, high-stakes settings.

**Broader Impact Concerns:**

NA`

**Claims And Evidence:**

Yes

**Claims Explanation:**

The submission provides clear and convincing evidence supporting its claims. The proposed methods are well motivated, grounded in established uncertainty estimation techniques, and evaluated through extensive experiments across diverse and realistic settings (e.g., distribution shift, cost sensitivity, limited expert data, and capacity constraints). The empirical results consistently demonstrate improved calibration and reduced misclassification cost compared to strong baselines, and the analyses (e.g., calibration error, OOD behavior) directly support the paper’s main claims. While the evaluation relies on synthetic experts and a single application domain, these limitations are acknowledged and do not undermine the validity of the presented evidence.

**Requested Changes:**

## Requested Changes

### Critical Changes (required for acceptance)

1. **Clarify generalization beyond the fraud detection domain**
   While the fraud detection setting is realistic and well motivated, all empirical evidence is drawn from a single application domain. The paper should more clearly justify why the proposed uncertainty-aware HAIC approaches are expected to generalize to other domains (e.g., healthcare, moderation, risk assessment), or explicitly scope the claims to similar tabular, alert-review settings. This clarification is important to properly contextualize the contribution.

2. **Stronger justification of the synthetic expert assumptions**
   Although the FiFAR/OpenL2D framework is more realistic than prior synthetic expert simulations, the paper should more clearly discuss which aspects of real human behavior are captured and which are not. In particular, limitations related to expert adaptation over time, correlated expert errors, and feedback loops should be discussed more explicitly, as these factors are central to human–AI collaboration.

3. **Sensitivity analysis of key hyperparameters**
   The performance of both proposed methods depends on important design choices (e.g., density model configuration, conformal coverage level optimization, weighting between predicted and average expert correctness). While some tuning details are relegated to the appendix, a clearer sensitivity analysis or robustness discussion in the main paper would strengthen confidence that improvements are not narrowly dependent on specific parameter settings.

---

### Non-Critical Changes (would strengthen the paper)

4. **Improve conceptual clarity between the two proposed approaches**
   While the paper presents Uncertainty-Aware L2D and Conformal Prediction for HAIC as complementary, the high-level guidance on *when* a practitioner should prefer one over the other could be made more explicit earlier in the paper. A short summary table or decision guideline would improve accessibility.

5. **Reduce technical overhead in the main exposition**
   Some sections (e.g., density-softmax integration and assignment optimization) are mathematically dense. Light restructuring, additional intuition, or a simplified running example could help readers less familiar with L2D or uncertainty estimation follow the core ideas without relying heavily on the appendix.

6. **Discuss computational and deployment costs more explicitly**
   Although the paper notes that density-softmax has low test-time overhead, the overall system includes normalizing flows, conformal prediction, and integer optimization. A brief discussion of training and inference cost trade-offs, and how these scale with problem size and number of experts, would improve practical relevance.

7. **Minor presentation improvements**

   * Add clearer captions or annotations to figures illustrating calibration and OOD behavior to make them more self-contained.
   * Ensure consistent terminology when referring to epistemic uncertainty, OOD data, and empty-set predictions across sections.

---

> ### Author Response · Authors · 2026-02-13
> **Response to Reviewer 5ow4**
>
> We thank the reviewer for the thorough review. We appreciate the engagement with both the conceptual framing and the technical components of the paper. Below, we respond to each requested change in turn.
> ### 1. Clarifying Generalization Beyond Fraud Detection
> We thank the reviewer for highlighting the importance of properly scoping our claims.
>
> In the revised manuscript, we clarify that our empirical evaluation is conducted in a tabular alert-review fraud detection setting and that our experimental claims are scoped accordingly. At the same time, we explicitly articulate why the proposed framework is expected to generalize beyond this domain.
>
> Specifically, we emphasize that our contribution is architectural rather than domain-specific. The framework consists of three modular components: (i) an uncertainty signal (epistemic proxy), (ii) routing/deferral policies, and (iii) capacity- and cost-constrained assignment optimization. These components are not tied to fraud detection and can be instantiated with alternative uncertainty estimators and prediction models appropriate to other modalities (e.g., healthcare or risk scoring).
>
> Accordingly, we have revised the Conclusions to state explicitly that empirical validation is restricted to tabular alert-review settings. We further clarify that extending the framework to other domains primarily requires selecting uncertainty proxies suited to the relevant data modality, and that validation in additional application domains remains important future work.
>
>
> ### 2. Stronger Justification of Synthetic Expert Assumptions
>
> We agree that a clearer discussion of which aspects of human decision-making are captured is important.
>
> In the revised manuscript, we expand the dataset discussion to explicitly state that the FiFAR/OpenL2D framework captures instance-dependent expert accuracy, heterogeneity across experts, and realistic inter- and intra-rater variability documented in high-stakes decision-making. At the same time, we explicitly acknowledge the limitations of the current setup, such as not modeling expert adaptation over time, and while expert behavior is influenced by the model’s score, other external factors affecting real-world decisions are not explicitly represented. To the best of our knowledge, this is currently the most realistic framework available for generating synthetic expert predictions in L2D research.
>
> We have added this clarification in the Dataset paragraph in Section 6, making clear that while the synthetic experts are more realistic than common class-based noise models, they remain an abstraction of real human decision processes.
>
>
> ### 3. Sensitivity Analysis of Key Hyperparameters
>
> We appreciate the request for a clearer robustness discussion in the main paper.
>
> In the revised version, we include a consolidated sensitivity discussion in the Results section, emphasizing two points:
>
> First, both proposed methods consistently outperform all baselines across approximately 300 experimental settings varying noise level, deferral rate, number of experts, and data availability. Because the two systems rely on different uncertainty mechanisms (density-softmax vs. conformal prediction) yet both yield consistent improvements, this suggests that the primary driver of performance is uncertainty-guided policy adaptation rather than any single design choice.
>
> Second, regarding specific hyperparameters:
>
> For the conformal approach, performance is most sensitive to the optimization of the coverage level α, which governs the balance between routing strategies. We report empirical evidence that the joint optimization procedure consistently identifies effective values across settings (see Results section and Appendix D).
>
> For density-softmax, there are no additional dedicated hyperparameters beyond those of the underlying ML models. The only design choice is the weighting between predicted and average expert correctness under high epistemic uncertainty; however, the effective weighting varies naturally across experimental settings due to differences in available labeled data per expert. The system is therefore not tuned to a fixed configuration.
>
> The remaining hyperparameters correspond to the underlying ML models (e.g., LightGBM, MLP, RealNVP) and are tuned using standard hyperparameter optimization procedures. Details of these tuning procedures are provided in the Appendix.
>
> This discussion is now included in the main Results section under a dedicated “Sensitivity and Robustness” subsection.

---

> ### Author Response · Authors · 2026-02-13
> **Response to Reviewer 5ow4 #2**
>
> ### 4. Clarifying When to Prefer Each Approach
>
> We thank the reviewer for suggesting clearer practitioner guidance.
>
> In the revised manuscript, we address this primarily by foregrounding our main conceptual contribution at the beginning of the paper. We now explicitly state that the core idea is uncertainty-driven policy switching: epistemic uncertainty is used as a control signal to determine which deferral policy should govern each instance. We also introduce a conceptual figure early in the Introduction that illustrates how the epistemic uncertainty measure guides the choice of routing strategy. This helps readers immediately grasp the system-level objective before encountering the technical details.
>
> With this framing in place, the two proposed approaches are more clearly positioned as alternative implementations of the same principle. The density-softmax method implements a smoother, calibration-oriented adjustment that is particularly effective under mild or gradual distribution shift, where uncertainty varies continuously. In contrast, the conformal prediction–based approach provides a more conservative, threshold-based mechanism that can be advantageous under more severe or more easily separable shift regimes. These distinctions are now discussed in the method sections and further illustrated in the Results section, where we show empirically how different shift regimes favor different operationalizations of uncertainty-driven policy adaptation.
>
> We believe this restructuring clarifies not only when a practitioner might prefer one approach over the other, but also that both instantiate the same underlying contribution: using epistemic uncertainty to guide deferral policy selection under distribution shift.
>
> ### 5. Reducing Technical Overhead and Improving Intuition
>
> We acknowledge that some sections of the paper are mathematically dense.
>
> In the revision, we focused on improving intuition and conceptual guidance rather than removing technical content. In particular, we strengthened the framing at the beginning of the paper by clearly highlighting the main contribution—uncertainty-driven policy switching—and introducing a conceptual figure that illustrates how epistemic uncertainty guides the choice of deferral policy. This provides readers with a high-level mental model before engaging with the formal details.
>
> We also added intuitive explanations preceding the density-softmax formulation and the joint assignment optimization. For density-softmax, we now frame it as a form of data-dependent temperature scaling that qualitatively approximates epistemic variance shrinkage in Bayesian models, helping distinguish it from an ad hoc modification. For the routing objective, we reinterpret the optimization as a constrained expected-risk maximization problem, providing decision-theoretic grounding before presenting the formal program.
>
> We did not remove technical details, as we believe they are necessary for precision and reproducibility. However, with the added conceptual framing and intuition, these sections should now be substantially easier to follow without sacrificing rigor.
>
> ### 6. Computational and Deployment Considerations
>
> We thank the reviewer for raising this point.
>
> In the revised manuscript, we add a paragraph discussing computational costs. For the uncertainty-aware L2D system, the additional component is the density model integrated into each OvA classifier. At inference, this requires a density evaluation and logit adjustment per classifier, scaling linearly with the number of experts, consistent with standard OvA L2D. At training time, the main additional cost is fitting one density model per expert-specific dataset.
>
> For the conformal prediction approach, the additional component is density-based conformal scoring. Training involves fitting the density estimator and computing empirical quantiles on a calibration set. At inference, this reduces to evaluating a density score and comparing it to a threshold (class-conditional in multiclass settings).
>
> This discussion is now included in the main Results section under a dedicated “Computational considerations.” subsection.
>
> ### 7. Minor Presentation Improvements
>
> We appreciate these suggestions. We have revised figure captions to make them more self-contained and clarified terminology throughout the paper. In particular, we now use epistemic uncertainty, OOD instances, and empty-set predictions carefully and consistently, emphasizing that they are related but not interchangeable concepts.

---

### Review · Reviewer_bDyr · 2026-01-29

**Summary Of Contributions:**

In my view, the paper elevates uncertainty from a model-level diagnostic to a system-level decision variable in human-AI collaboration under distribution shift. It demonstrates that robust human-AI collaboration under shift is fundamentally a policy selection problem, not merely a better prediction problem, and uncertainty is the correct control signal for that selection.

Uncertainty is explicitly operationalized to decide which decision policy to trust—fine-grained learning-to-defer versus conservative rejection—depending on epistemic conditions.
This resolves a long-standing tension in HAIC between expressiveness (L2D) and robustness (ReL) under distribution shift. The work shows that the distribution shift should not be handled by a single monolithic policy. Instead, it proposes a principled policy switching mechanism driven by uncertainty via density-aware calibration or conformal emptiness, which is both interpretable and deployable.

I think the paper is impactful beyond fraud detection or the specific architectures used.

**Audience:**

Yes

**Audience Explanation:**

1. This paper directly addresses how uncertainty should be used, not merely estimated, in the presence of distribution shift.

2. The paper goes beyond standard classification or uncertainty benchmarking and studies decision-making systems that combine models, humans, and constraints.

3. Although evaluated in a fraud detection setting, the core findings—such as uncertainty-driven policy switching and the tradeoff between expressive routing and conservative rejection—generalize to many high-stakes domains.

**Broader Impact Concerns:**

The proposed methods implicitly determine when decisions are automated and when they are deferred to human experts, thereby influencing human workload, responsibility allocation, and accountability. Systematic deferral of certain subpopulations—particularly those associated with low-density or out-of-distribution regions—could lead to uneven human scrutiny or delays, potentially introducing fairness concerns if such regions correlate with sensitive attributes.

**Claims And Evidence:**

Yes

**Claims Explanation:**

- The experiments clearly demonstrate that uncertainty-aware routing improves robustness under distribution shift compared to standard Learning-to-Defer and pure Rejection Learning baselines.
- The controlled noise-injection setup shows consistent trends: density-aware calibration improves probability calibration (ECE), and conformal-based rejection becomes increasingly effective as shift severity grows.
- The ablation-style comparisons (DS vs. CP vs. L2D vs. ReL) convincingly support the claim that no single policy dominates across all shift regimes, motivating uncertainty-driven policy selection.

**Requested Changes:**

- The paper frequently equates low density estimates or conformal empty prediction sets with epistemic uncertainty, but this identification is largely empirical and not formally justified. In complex or multimodal distributions, low likelihood does not necessarily imply epistemic uncertainty, and conformal emptiness under distribution shift does not come with theoretical guarantees. Clarify that density and conformal emptiness are used as proxies for epistemic uncertainty rather than exact characterizations. Please add a short discussion on when this assumption may break (e.g., conditional shift or sparse in-distribution regions).

- The paper’s most valuable insight—that robust human–AI collaboration under distribution shift requires uncertainty-driven policy switching rather than a single deferral strategy—is not clearly highlighted as a central contribution. As a result, the work may appear as a collection of techniques rather than a coherent conceptual advance. I suggest providing a conceptual illustration of different regimes of distribution shift and the corresponding optimal policies.

- The use of conformal prediction set emptiness as a trigger for conservative deferral implicitly treats emptiness as a reliable indicator of epistemic shift. However, conformal prediction guarantees coverage only under exchangeability, and its behavior under distribution shift is not theoretically grounded.

- The joint optimization over routing decisions, expert capacity, and conformal thresholds is a strong systems contribution, but the objective function is presented in an engineering-centric manner without a clear decision-theoretic interpretation.

- The combination of density estimation, conformal prediction, learning-to-defer, and constrained optimization can give the impression of a method stack rather than a unified framework, potentially obscuring the paper’s central message.

- Provide an interpretive justification for density-softmax, for example by framing it as a data-dependent temperature scaling mechanism or as an approximation to epistemic variance shrinkage, even a qualitative theoretical interpretation would help distinguish it from an ad-hoc engineering choice.

---

> ### Author Response · Authors · 2026-02-13
> **Response to Reviewer bDyr**
>
> We thank the reviewer for the thorough review. We appreciate the engagement with both the conceptual framing and the technical components of the paper. Below, we respond to each requested change in turn.
>
> ### 1. Density and Conformal Emptiness as Proxies for Epistemic Uncertainty
>
> We agree that low density estimates and conformal empty prediction sets should not be interpreted as formal or exact characterizations of epistemic uncertainty. Our intention was to use density scores and conformal emptiness as empirical proxies for epistemic uncertainty, following prior work (see references [1,2,3] below) that links low support under the training distribution with increased epistemic uncertainty in practice.
>
> In the revised manuscript, we have clarified that:
> - Density-based scores and conformal empty sets are proxies for epistemic uncertainty rather than theoretically exact quantities.
> - Conformal prediction guarantees hold under exchangeability and do not extend to arbitrary distribution shifts.
> - The relationship between low density and epistemic uncertainty may break under certain forms of shift, such as complex or multimodal distributions, or under conditional shifts that preserve density while altering label structure
>
> We also explicitly state that our claims are empirical. In regimes where the distribution shift corresponds to a movement into low-support regions, the proxy performs well, as demonstrated in our experiments; however, this correspondence must be validated in other distributional settings.
>
> These clarifications have been added to Section 4 (second paragraph) and Section 7 (sixth paragraph) of the revised manuscript.
>
> [1] Eyke Hüllermeier and Willem Waegeman. Aleatoric and epistemic uncertainty in machine learning: an
> introduction to concepts and methods. Machine Learning, 110(3):457–506, March 2021. ISSN 1573-0565.
> doi: 10.1007/s10994-021-05946-3. URL http://dx.doi.org/10.1007/s10994-021-05946-3.
>
> [2] Yotam Hechtlinger, Barnabás Póczos, and Larry A. Wasserman. Cautious Deep Learning.
> abs/1805.09460, 2018. URL http://arxiv.org/abs/1805.09460.
>
> [3] Soundouss Messoudi, Sylvain Rousseau, and Sébastien Destercke. Deep conformal prediction for robust
> models. In Marie-Jeanne Lesot, Susana M. Vieira, Marek Z. Reformat, João Paulo Carvalho, Anna Wilbik,
> Bernadette Bouchon-Meunier, and Ronald R. Yager (eds.), Information Processing and Management of
> Uncertainty in Knowledge-Based Systems - 18th International Conference, IPMU 2020, Lisbon, Portugal,
> June 15-19, 2020, Proceedings, Part I, volume 1237 of Communications in Computer and Information
> Science, pp. 528–540. Springer, 2020. doi: 10.1007/978-3-030-50146-4\_39. URL https://doi.org/10.
> 1007/978-3-030-50146-4_39.
>
>
> ### 2. Highlighting the Central Conceptual Contribution (Policy Switching)
>
> We appreciate the suggestion that the central conceptual contribution of uncertainty-driven policy switching should be emphasized more clearly.
>
> In the revision, we reframed the Introduction and contribution statements to explicitly position the paper around the central claim that robust human–AI collaboration under distribution shift requires uncertainty-driven policy selection. Specifically, we clarify that distribution shift should not be handled by a single deferral mechanism, but by switching between routing policies according to epistemic uncertainty. We added a conceptual illustration describing how epistemic uncertainty affects the choice of the optimal deferral policy.
>
> These clarifications are in the Introduction (fifth paragraph, contributions list and Figure 1) and in the Conclusions.
>
> This restructuring aims to make clear that the proposed methods instantiate a broader contribution, rather than constituting a collection of loosely connected techniques.
>
>
> ### 3. Conformal Emptiness Under Distribution Shift
>
> We agree that conformal prediction’s guarantees rely on exchangeability and do not extend to arbitrary distribution shifts. We have revised the manuscript to explicitly state the exchangeability assumption and its implications, and clarify that empty-set predictions are used as a conservative deferral trigger rather than as a theoretically guaranteed detector of epistemic shift. Furthermore, we discuss scenarios in which conformal emptiness may not reliably indicate epistemic uncertainty.
>
> These clarifications have been added to Section 4 (second paragraph) and Section 7 (sixth paragraph) of the revised manuscript.

---

> ### Author Response · Authors · 2026-02-13
> **Response to Reviewer bDyr #2**
>
> ### 4. Decision-Theoretic Interpretation of the Optimization
>
> We appreciate the suggestion to provide a clearer decision-theoretic interpretation of the joint optimization over routing, capacity, and conformal thresholds.
>
> In the revised version, we clarify that the objective corresponds to maximizing expected utility under uncertainty. We interpret predicted correctness probabilities as instance-level expected rewards, and describe the routing decision as selecting an action from a constrained decision space defined by capacity and cost considerations.
>
> These clarifications have been added to the capacity constraints paragraph in Section 5 of the revised manuscript.
>
> This reframing situates the optimization within a principled decision-theoretic framework rather than an engineering heuristic.
>
>
> ### 5. Unified Framework vs. “Method Stack”
>
>
> We understand the concern that the combination of density estimation, conformal prediction, L2D, and constrained optimization may appear as a stacked set of techniques.
>
> To address this, we have clarified that both proposed systems share a common structural principle: an uncertainty signal, routing/deferral policies, and capacity-constrained assignment, highlighting this as the central conceptual contribution. The two implementations (density-softmax and conformal prediction) instantiate different methods for generating the uncertainty signal, but the underlying systems perspective is consistent.
>
> These clarifications are in the Introduction (fifth paragraph, contributions list and Figure 1), in the Conclusion, and in the final paragraph of the Results section.
>
> This aims to reinforce the coherence of the framework and that the central contribution is not the individual components, but the integration of uncertainty as the variable for policy selection.
>
>
> ### 6. Interpretive Justification for Density-Softmax
>
> We agree that density-softmax benefits from a clearer interpretive framing.
>
> In the revised manuscript, we provide a qualitative interpretation of density-softmax as:
> - A data-dependent confidence modulation method.
> - An approximation to epistemic variance shrinkage in Bayesian models, where uncertainty over parameters reduces predictive confidence away from observed data.
> - We clarify that, unlike Bayesian inference, density-softmax does not marginalize over parameter distributions, but instead deterministically attenuates logits in low-support regions to approximate similar behavior.
>
> These clarifications have been added to the density-softmax paragraph in Section 3 of the revised manuscript.
>
>
>
> ### 7. Broader Impact and Fairness Concerns
>
> We appreciate the reviewer’s thoughtful comments regarding workload distribution, responsibility allocation, and fairness implications.
>
> In the revised manuscript, we explicitly acknowledge that uncertainty-driven deferral policies influence human workload and responsibility allocation. We note that if low-density regions correlate with sensitive attributes, systematic deferral may concentrate scrutiny or delays on specific subpopulations. Therefore we highlight the need for subgroup-level analysis of deferral patterns and fairness-aware routing constraints in deployment in future work.
>
> We add this discussion in the conclusion, when discussing future work.
>
> We believe this addition strengthens the broader impact discussion and clarifies that deployment requires fairness monitoring.

---

### Review · Reviewer_LN8K · 2026-02-09

**Summary Of Contributions:**

The paper studies human–AI decision pipelines under distribution shift with limited human capacity and asymmetric costs. It proposes two uncertainty-aware routing methods: (i) a density-based logit scaling (“density-softmax”) to reduce overconfident predictions in low-density regions of a learned representation, and (ii) a density-based conformal-style set predictor coupled with a capacity-constrained assignment optimizer to decide when to defer to humans. Experiments on a fraud-alert dataset with simulated shift suggest improved calibration and allocation relative to uncertainty-agnostic baselines.

**Audience:**

Yes

**Audience Explanation:**

Yes. The operational focus on capacity constraints and cost-sensitive routing is practically relevant.

**Claims And Evidence:**

Yes

**Claims Explanation:**

I am not an expert in learning-to-defer and I may miss relevant related work; my comments focus on correctness and statistical clarity.

My main concern is an apparent error in the stated Bayes-optimal deferral rule. The paper compares terms involving $\max _y \mathbb{P}\left(h^\*(x)=y \mid x\right)$ where $h^\*(x)$ is defined as an argmax classifier. As written, $h^\*(x)$ is deterministic given $x$, so this probability is 0 or 1 and the maximum is identically 1 , making the criterion degenerate. I believe the intended quantity is the model's probability of being correct, such as $\max _y \mathbb{P}(Y=y \mid x)$, and similarly for experts $\mathbb{P}\left(Y=m_j(x) \mid x\right.$ ), or a calibrated correctness estimate used in standard L2D formulations. This needs correction or a careful re-derivation.

That being said, the paper’s main strengths are its clear focus on a realistic deployment constraint and its attempt to build uncertainty-awareness directly into the routing mechanism rather than treating calibration as an afterthought. The proposed implementations are also fairly concrete, making the ideas easy to reproduce and potentially useful beyond the specific fraud-alert setting.

**Requested Changes:**

Correct and clearly explain the Bayes-optimal deferral rule (what random variable the probability is taken over, and what quantity is being compared).

---

> ### Author Response · Authors · 2026-02-13
> **Response to Reviewer LN8K**
>
> Thank you for the thorough review, and for identifying the issue in the stated Bayes-optimal deferral rule.
>
> You are correct that, as originally written, the criterion is degenerate, since $h^*(x)$ is deterministic given $x$. The relevant quantity is instead the probability that each decision-maker (the classifier or a given expert) is correct with respect to the true label.
> We have revised the manuscript to clarify the underlying random variable and to state the Bayes-optimal solution explicitly in terms of correctness probabilities. The corrected L2D formulation is now in the revised submission. This revision makes explicit that the comparison is between the classifier’s probability of being correct and the corresponding probabilities for each expert, in line with standard L2D formulations.
>
> Importantly, our implementation and empirical evaluation already rely on estimated probabilities of correctness for both the classifier and the experts. Therefore, this correction is notational and clarificatory in nature and does not affect the methodology or the reported results.

---

> > ### Comment · Reviewer_LN8K · 2026-02-14
> > **Response to the authors' revision**
> >
> > Thanks for the correction! I'm happy with the revision.

---

### Review · Reviewer_d3bB · 2026-02-20

**Summary Of Contributions:**

This paper studies the problem of collaborative decision-making using both an AI system and human experts. The task is to decide how to allocate decisions between the AI system and the different experts to minimize the expected misclassification cost while respecting expert capacity constraints.

The authors describe two improvements to current methods for this problem that both focus on improving robustness to OOD data:
* They incorporate out-of-distribution uncertainty into learning to defer (L2D) by scaling logits according to a normalizing flow likelihood.
* They propose using conformal prediction to decide when to defer to experts, which also leverages OOD detection.

**Audience:**

Yes

**Audience Explanation:**

I am not an expert in this particular problem, but it seems important and there is a community around it. I am convinced that incorporating OOD detection addresses an important limitation of previous methods (assuming that the authors have represented the previous literature well—I did not carefully check for novelty).

**Broader Impact Concerns:**

No concerns.

**Claims And Evidence:**

Yes

**Claims Explanation:**

In general the results suggest that the two proposed methods reduce misclassification cost compared to baselines. The methods are well-motivated and seem reasonable.

**Requested Changes:**

I think the main weakness of the paper is the organization/writing. I found that reading through the paper, there was not enough signposting to clearly understand what the main content of each section was going to be; the authors often dive right into technical details without first introducing what they are going to discuss. It was difficult at times to tell which parts of each section were describing previous methods from the literature and which parts were describing the authors' novel contributions. I also found it a bit confusing that the authors propose two methods—at first I thought these were going to be combined, but my understanding is that they are just separate ideas and are evaluated separately. Overall, I think improving the organization and clarity would help the paper have more impact.

---

> ### Author Response · Authors · 2026-02-25
> **Response to Reviewer d3bB**
>
> We thank the reviewer for this thoughtful feedback. In the revised manuscript, we first improved the organization and signposting throughout the paper. In particular, the beginnings of Sections 3 and 4 now provide clear high-level overviews of each method, outlining their objective, their relationship to existing approaches, and the specific novelty of our contributions before introducing technical details. This restructuring is intended to make it immediately clear what are the contributions of our work.
>
> We also strengthened the framing in the Introduction and contribution statements to explicitly position the paper around the central claim that robust human–AI collaboration under distribution shift requires uncertainty-driven policy selection. Importantly, we clarify early on that we implement this idea through two distinct uncertainty-aware systems, which instantiate the same conceptual framework via different mechanisms and are evaluated separately. Together, these changes aim to improve the structure and organization of the paper.

---

### Decision · Action_Editor_YGVF · 2026-02-27

**Recommendation:** Accept as is

**Audience:**

Yes

**Audience Explanation:**

The HAI community, particularly those with interest in learning to defer, may find this work interesting.

**Claims And Evidence:**

Yes

**Claims Explanation:**

After revision, authors have strengthened their piece, provided additional evidence for claims, and improved clarity to the satisfaction of the reviewers and myself.